# Nutrient deprivation alters the rate of COPII subunit recruitment at ER subdomains to tune secretory protein transport

William Kasberg [1], Peter Luong[1], Kevin A. Swift[1] & Anjon Audhya [1] ✉

Co-assembly of the multilayered coat protein complex II (COPII) with the Sar1 GTPase at subdomains of the endoplasmic reticulum (ER) enables secretory cargoes to be concentrated efficiently within nascent transport intermediates, which subsequently deliver their contents to ER-Golgi intermediate compartments. Here, we define the spatiotemporal accumulation of native COPII subunits and secretory cargoes at ER subdomains under differing nutrient availability conditions using a combination of CRISPR/Cas9-mediated genome editing and live cell imaging. Our findings demonstrate that the rate of inner COPII coat recruitment serves as a determinant for the pace of cargo export, irrespective of COPII subunit expression levels. Moreover, increasing inner COPII coat recruitment kinetics is sufficient to rescue cargo trafficking deficits caused by acute nutrient limitation. Our findings are consistent with a model in which the rate of inner COPII coat addition acts as an important control point to regulate cargo export from the ER.

The vast majority of newly synthesized secretory cargoes are translated into the lumen of the endoplasmic reticulum (ER) and recognized by a variety of receptors that promote their packaging into transport intermediates[1]. The prototypical cargo receptor Sec24[2–4] functions as part of a multi-subunit complex of coat proteins collectively known as COPII[5–7], which assembles immediately adjacent to ribosome-free ER subdomains (transitional elements of the ER) harboring the scaffolding protein Sec16a[8–13], members of the Tango1/cTAGE family[14–17], and the guanine nucleotide exchange factor Sec12[13,18]. Based on numerous studies over the past several decades, a detailed model has evolved to describe the earliest steps of COPII-mediated trafficking. Specifically, Sec12 first promotes GTP loading onto the small GTPase Sar1[18–21], enabling stable penetration of its amino-terminal amphipathic helix into the lipid bilayer to induce membrane tubulation[22–27] while simultaneously directing the recruitment and organization of Sec23-Sec24 heterodimers to form the inner layer of the COPII coat complex[28–30]. Both Sec16a[8,30,31] and Tango1/cTAGE[15,32] also interact directly with Sec23-Sec24 and have been suggested to increase their local concentration, thereby stimulating coat formation. Sec16a additionally associates with Sec13-Sec31 heterotetramers[33], which co-assemble around the inner coat to form an outer COPII cage[29,34]. This dual-layered coat complex has been proposed to sculpt membrane tubules generated by activated Sar1 to form nascent cargo-laden transport intermediates[25,26,29,35]. In a manner that is likely dependent on Sar1 GTP hydrolysis[22,23,26,36], these intermediates undergo maturation and subsequently deliver their contents to distinct ER-Golgi intermediate compartments (ERGIC)[37–39] or an interwoven tubular network that is connected to the ER[40–42], which is facilitated by the Sec23-binding protein TFG[43–45].

The Sar1 GTPase cycle has also been implicated in controlling the timing of COPII coat disassembly[23,26,36,37], which is required for cargo delivery at acceptor membranes[46,47]. Somewhat paradoxically, the inner coat protein Sec23 functions as the guanine nucleotide activating protein (GAP) for Sar1[48–50], inserting an arginine finger into its active site[28,29,34], which is further stimulated by the outer coat subunit Sec31[36,49]. Therefore, regulatory mechanisms must exist to tightly control Sec23 GAP activity during transport intermediate formation, both to prevent premature COPII coat disassembly, but also to maintain a rapid rate of anterograde cargo transport at the ER/ERGIC interface. Several cell signaling pathways have been suggested to function in this context. Governed by nutrient availability[51–58], various forms of cell stress[58], and the presence of growth factors[59], COPII

---

[1]Department of Biomolecular Chemistry, University of Wisconsin School of Medicine and Public Health, Madison, WI 53706, USA. ✉e-mail: audhya@wisc.edu

subunits and multiple COPII regulatory factors undergo post-translational modifications, including phosphorylation[51,52,59–61], glycosylation[62,63], and ubiquitylation[64], leading to alterations in their local concentrations at sites of transport intermediate formation, which in turn tunes the kinetics of cargo export. Additionally, their total cellular levels are further controlled by events that alter protein stability[53,57] or gene expression[58,65]. Perhaps most notably, acute serum and amino acid deprivation have been demonstrated to alter the phosphorylation state of several COPII subunits as well as Sec16a, reducing their levels at ER subdomains and impairing the rate of COPII-mediated cargo transport[51,59]. Similarly, reduced activity of the IRE1 branch of the unfolded protein response pathway resulting from prolonged absence of nutrients was shown to downregulate the expression of numerous factors involved in COPII-mediated trafficking, including Sec16a and isoforms of Sec23, Sec24, and Sec31[58].

Despite our expansive understanding of the regulatory systems that control post-translational modifications on COPII subunits and their levels of expression under various conditions, the dynamics of COPII recruitment to ER subdomains remain poorly defined. This is in part due to the relative dearth of live cell imaging studies examining native COPII subunits. Instead, the majority of published work has focused on fixed cell analysis or the use of ectopic overexpression, which can alter protein dynamics, to determine COPII distribution. Here, we leverage CRISPR/Cas9-mediated genome editing to establish a series of human cell lines that endogenously express HaloTag fusions to several COPII subunits, as well as key COPII regulatory factors. In combination with lattice light-sheet imaging, we define the rates of COPII subunit incorporation at ER subdomains and demonstrate that acute nutrient deprivation slows the kinetics of Sec23 addition. This defect is accompanied by a diminished rate of anterograde cargo transport, consistent with previous work[51,52,58,59]. However, by artificially increasing the rate of Sec23 incorporation, we show that cargo trafficking defects induced by short-term nutrient deprivation can be rescued. Taken together, our findings are most consistent with a model in which the rate of inner COPII coat recruitment dictates the kinetics of secretory cargo packaging and export from the ER.

## Results

### COPII subunits endogenously appended with HaloTag function normally in secretory protein trafficking

Although COPII-mediated trafficking has been successfully reconstituted in vitro[6], relatively little is known about the regulatory mechanisms that govern the rate of cargo egress from the ER in mammalian cells. To address this issue, we first leveraged CRISPR/Cas9 genome editing to append the modular HaloTag on several factors implicated directly in regulating the anterograde transport of newly synthesized secretory proteins. In particular, we focused on the scaffolding protein Sec16a, which marks subdomains of the ER from which secretory cargoes emerge, components of the inner and outer COPII coat, Sec23a and Sec31a respectively, and the Sec23-binding protein TFG, which has been suggested to organize COPII transport intermediates at the ER/ERGIC interface (Supplementary Fig. 1a). HaloTag was selected due to its unique ability to rapidly and irreversibly bind to cell permeable ligands that are coupled to bright and photostable fluorescent dyes, making them ideal for live cell imaging[66]. CRISPR/Cas9 editing was conducted using immortalized human Retinal Pigment Epithelial (RPE1) cells, which exhibit a stable diploid karyotype, and multiple clones expressing each fusion protein in the absence of overexpression were identified using immunoblot analysis and sequencing. Individual clones expressing native levels of HaloTag-Sec31a and HaloTag-TFG in a homozygous manner were selected for further study, as was a clone expressing HaloTag-Sec16a from a single allele, since we were unable to identify any that were edited homozygously (Fig. 1a and Supplementary Fig. 1b, c). We additionally took

advantage of RPE1 cells natively expressing HaloTag-Sec23a, which we described previously[67].

To verify the functionality of each fusion protein, we first examined the proliferation rates of the edited cell lines, since ongoing COPII-mediated trafficking is essential for viability[7]. With the exception of cells expressing HaloTag-Sec16a, which grew significantly slower, all other cell lines exhibited a doubling time that was similar to control, unedited RPE1 cells (Supplementary Fig. 1d). We next determined the distributions of each fusion protein using confocal and super-resolution fluorescence microscopy, which showed that they all localized appropriately (350–400 structures per cell on average), juxtaposed to sites decorated with antibodies directed against endogenous Sec24a, another component of the COPII coat (Fig. 1b, c and Supplementary Fig. 1e). Linescan analysis further confirmed these findings, strongly suggesting that the addition of HaloTag does not interfere with targeting of the fusion proteins (Fig. 1c). Importantly, organization of the early secretory pathway was not impacted by their expression, as indicated by normal distributions of ER, ERGIC, and Golgi markers, as compared to control RPE1 cells (Supplementary Fig. 2).

To assess whether the rate of anterograde trafficking was affected in cells expressing HaloTag fusion proteins, we conducted two distinct synchronized cargo release assays[68,69]. In the first case, an ER export sorting motif appended onto DsRed (ss-DsRed) was expressed as a fusion to a mutant form of the FK506-binding protein (FKBP) that causes its aggregation in the ER lumen. Only upon solubilization mediated by the addition of synthetic ligand of FKBP (SLF) does ss-DsRed undergo packaging into COPII transport intermediates for export from the ER[68]. In a second approach, the human invariant chain of the major histocompatibility complex fused to streptavidin, which is translated into the ER lumen but unable to leave, was co-expressed with a fragment of the Golgi resident enzyme Mannosidase II fused to streptavidin binding protein and GFP (ManII-SBP-GFP), restricting it to the ER lumen until the addition of biotin enables dissociation and subsequent export[69]. Live cell confocal imaging was used to monitor movement of ss-DsRed and ManII-SBP-GFP following release in control and CRISPR/Cas9-edited cell lines. We found that neither cargo was affected by the expression of HaloTag fusion proteins, with the exception of a brief delay in ManII-SBP-GFP trafficking in cells expressing HaloTag-Sec16a (Fig. 1d and Supplementary Fig. 3).

The altered growth and cargo trafficking rates associated with cells expressing HaloTag-Sec16a raised some concerns regarding their utility for our studies. We thereafter analyzed the recovery kinetics of HaloTag-Sec16a following photobleaching to determine whether its mobility in cells had been impacted. Similar to previous work[70], approximately 60% of HaloTag-Sec16a was highly mobile, with a rapid half-time to maximal recovery of 3.1 +/− 0.9 s (Fig. 1e). Taken together, our findings strongly suggest that Sec23a, Sec31a, and TFG retain functionality when appended with the HaloTag, while Sec16a function may be partially perturbed under this condition, although its dynamics do not appear to be dramatically affected.

### Components involved in COPII-mediated trafficking exhibit distinct recruitment kinetics at ER subdomains

The establishment of cell lines natively expressing tagged isoforms of key regulators that direct anterograde secretory protein transport affords the unique opportunity to define the manner in which these factors accumulate at ER subdomains to influence cargo export. For these studies, we leveraged lattice light-sheet microscopy, an approach that provides near diffraction-limited resolution and collects full cell volumes with high speed and low levels of phototoxicity as compared to other forms of live cell imaging. Each cell line was labeled using fluorogenic HaloTag ligands and imaged continuously for 10 min. Consistent with previous findings[71,72], we identified both long-lived sites, which persisted beyond the confines of our imaging

window, and shorter-lived sites that harbored each marker (Fig. 2a and Supplementary Movies 1–4). Notably, in all cell lines examined, site longevity was correlated strongly with intensity (r values between -0.30 and -0.42) and more weakly with diameter (r values between -0.09 and -0.21), indicating that the longer-lived structures were generally larger and more intense, potentially representing multiple, closely juxtaposed COPII budding events (Supplementary Fig. 4). Therefore, to help ensure that our analysis would focus only the

dynamics of individual ER subdomains, we excluded sites that continued to be present at the termination of imaging datasets. Additionally, those that were already present at the start of imaging were also excluded, as it was impossible to determine when they had initially formed.

Using Imaris software, we were then able to measure the lengths of their individual lifetimes, which indicated that Sec16a exhibits significantly more stability at ER subdomains (81.5 +/− 3.3 s on average) as

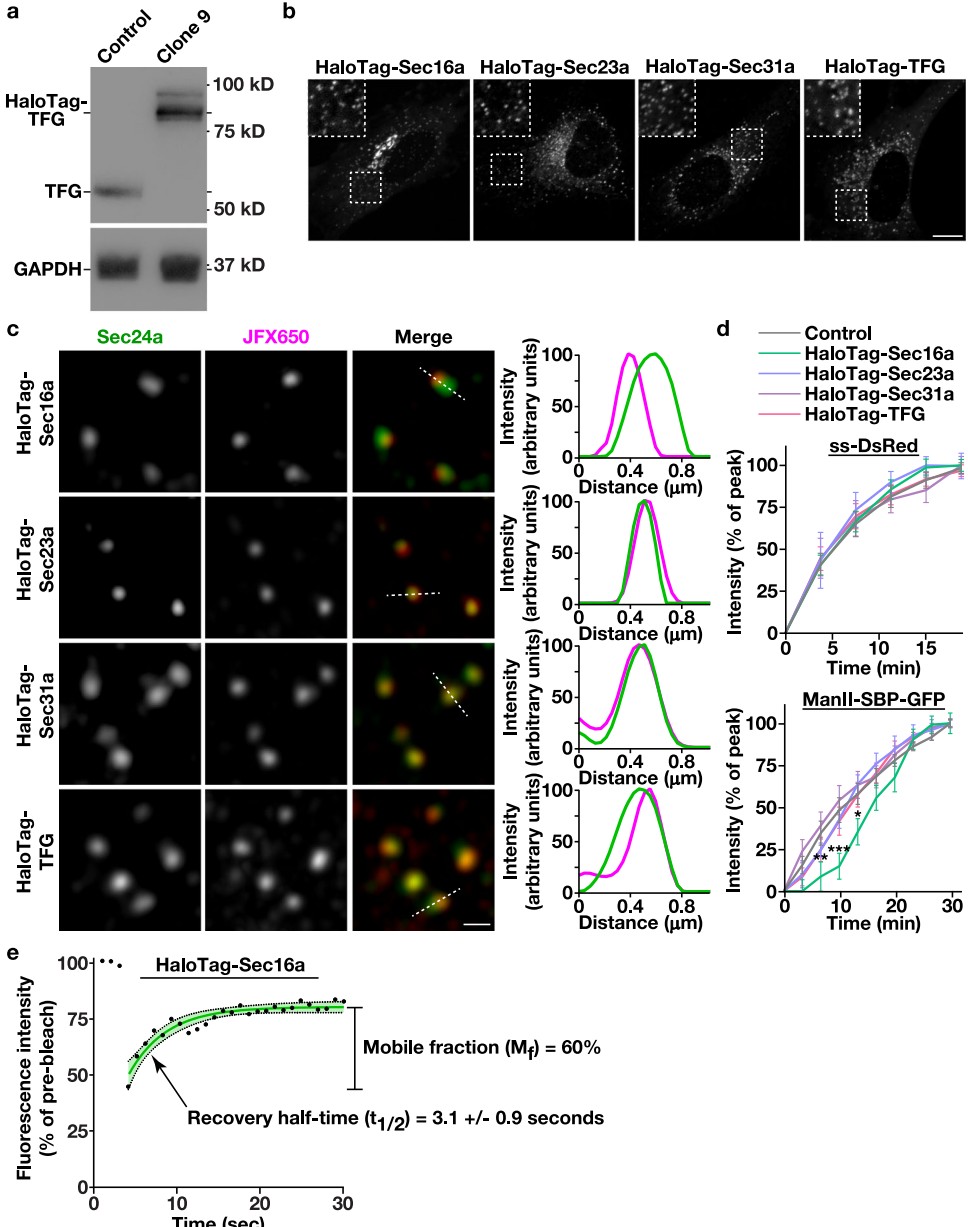

**Fig. 1 | Engineering human cell lines to study native COPII dynamics.**
**a** Representative immunoblots (n = 3) of extracts generated from control and clonal CRISPR/Cas9 edited cells using antibodies directed against TFG (top) and GAPDH (bottom). **b** Representative confocal images of cell lines natively expressing HaloTag fusion proteins after labeling with JFX650-HaloTag ligand are shown. Insets show 3x zoomed regions (boxed). Bar, 10 μm. **c** Representative super resolution images of fixed cells expressing HaloTag fusion proteins labeled with the JFX650-HaloTag ligand (magenta) and co-stained using antibodies directed against Sec24a (green). Color-coded linescans highlighting their relative localizations are shown (right). Bar, 500 nm. **d** Confocal imaging of edited cell lines (HaloTag-Sec16a, green; HaloTag-Sec23a, light blue; HaloTag-Sec31a, purple; HaloTag-TFG, salmon) co-expressing either ss-DsRed (top) or ManII-SBP-GFP

(bottom) was used to monitor their synchronous release from the ER. Based on fluorescence intensity, the percentage of each cargo present within the perinuclear region relative to its maximal accumulation there was quantified over time. Error bars represent mean +/- SEM (n = 20 cells each; 3 biological replicates each). ***p = 0.0003, **p = 0.0082, and *p = 0.0260, as calculated using a two-way ANOVA and Dunnett's multiple comparison test. **e** Confocal microscopy was used to monitor the fluorescence recovery of HaloTag-Sec16a labeled with JFX650-HaloTag ligand after partial photobleaching (n = 20; 3 biological replicates). Error, as displayed by green-colored bands around the smoothed tread line, represents mean +/- SEM. Based on the recovery curve, the mobile fraction (M$_f$) and half-time of recovery (t$_{1/2}$) were determined. Source data are provided as a Source Data file.

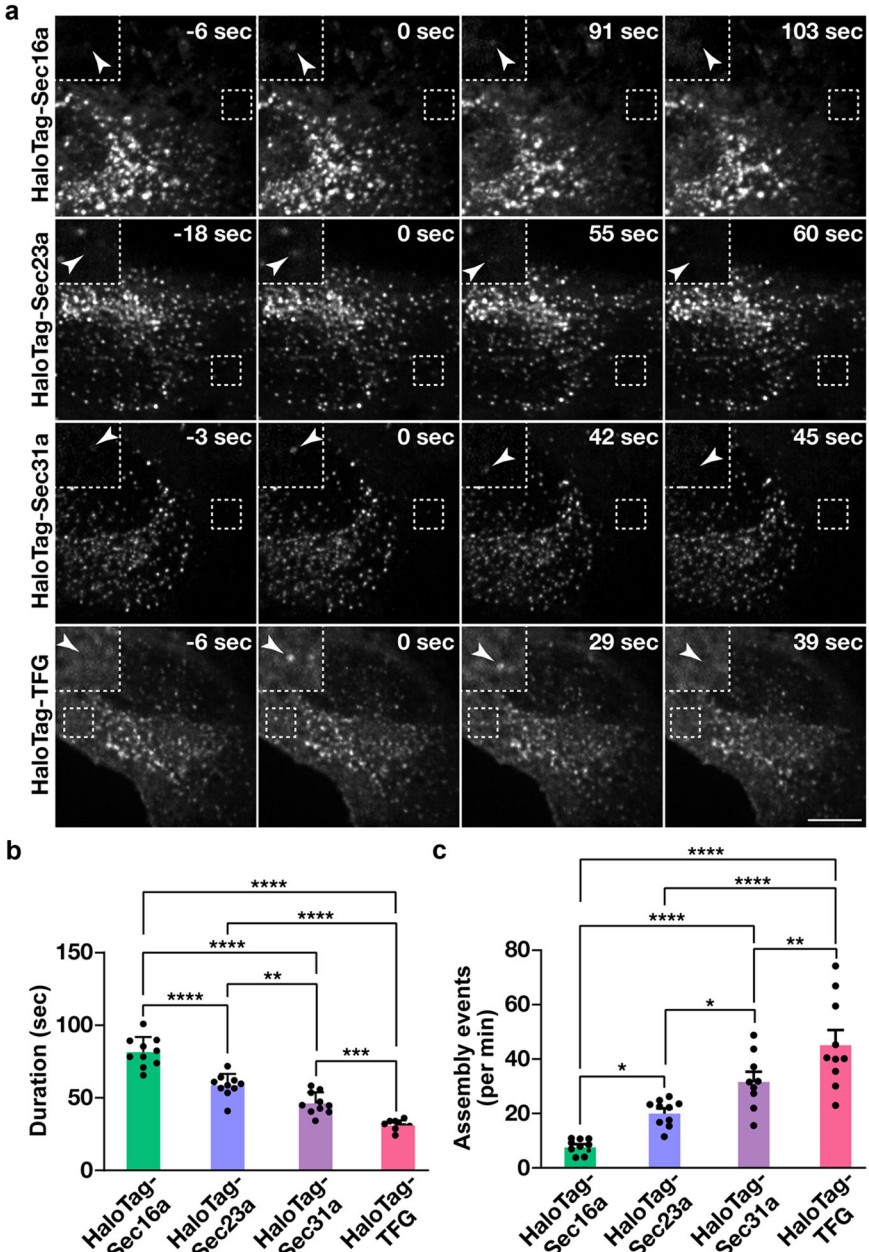

**Fig. 2 | COPII dynamics under nutrient replete conditions. a** Representative timelapse series showing the distributions of HaloTag fusion proteins following labeling with the JFX650-HaloTag ligand (imaged using lattice light-sheet microscopy). Insets (3x zoom) highlight structures indicated by arrowheads that assemble and disassemble during the imaging window. Bar, 10 μm. **b**, **c** Quantification of the average duration of each HaloTag fusion protein (HaloTag-Sec16a, green; HaloTag-Sec23a, light blue; HaloTag-Sec31a, purple; HaloTag-TFG, salmon) at an ER subdomain (**b**) and the number of new structures that assemble each minute (**c**). Error bars represent mean +/- SEM (n = 10 cells each; 3 biological replicates; more than 3000 tracked particles each). ****p < 0.0001, ***p < 0.001, **p < 0.01, and *p < 0.05, as calculated using a one-way ANOVA and Tukey post hoc test. Specific p values are provided in the Source Data file.

compared to either COPII subunit or TFG (Fig. 2b). Additionally, we found that the inner COPII coat persists for a longer period of time (58.5 +/− 2.5 s) than the outer coat (46.24 +/− 2.4 s), while TFG acts most transiently (30.9 +/− 1.1 s) (Fig. 2b). By analyzing the frequency distribution of their lifetimes, we found that their durations were best modeled by two phase exponential decay (Supplementary Fig. 5a–d), with the time constants associated with the slower phase being similar to the average lifetimes of each factor (Supplementary Fig. 5e, f). We also measured how often new sites appeared during the imaging period and found that TFG-positive structures, while shortest lived at ER subdomains, exhibited the highest frequency of de novo formation (Fig. 2c). In contrast, formation of new Sec16a-labeled sites was relatively infrequent, approximately 6-fold less as compared to TFG (Fig. 2c).

We next quantified the intensities of individual structures as they formed (Fig. 3a), yielding averaged assembly curves for each of the HaloTag fusion proteins (Fig. 3b and Supplementary Fig. 6a–d). Each had shared characteristics, including an early period of increasing fluorescence intensity, which was followed by a plateau phase. As the number of sites that could be analyzed diminished due to their dissolution, the plateau phases became more stochastic with time (Supplementary Fig. 6a–d). We therefore focused specifically on their formation and calculated both their instantaneous rates of fluorescence change during the first 50 s (Fig. 3c) and their associated rate constants (Supplementary Fig. 6e). These data demonstrated that Sec16a sites form most rapidly (i.e., approximately 1.8-fold faster than HaloTag-Sec23a), but they transition quickly toward a slower, more

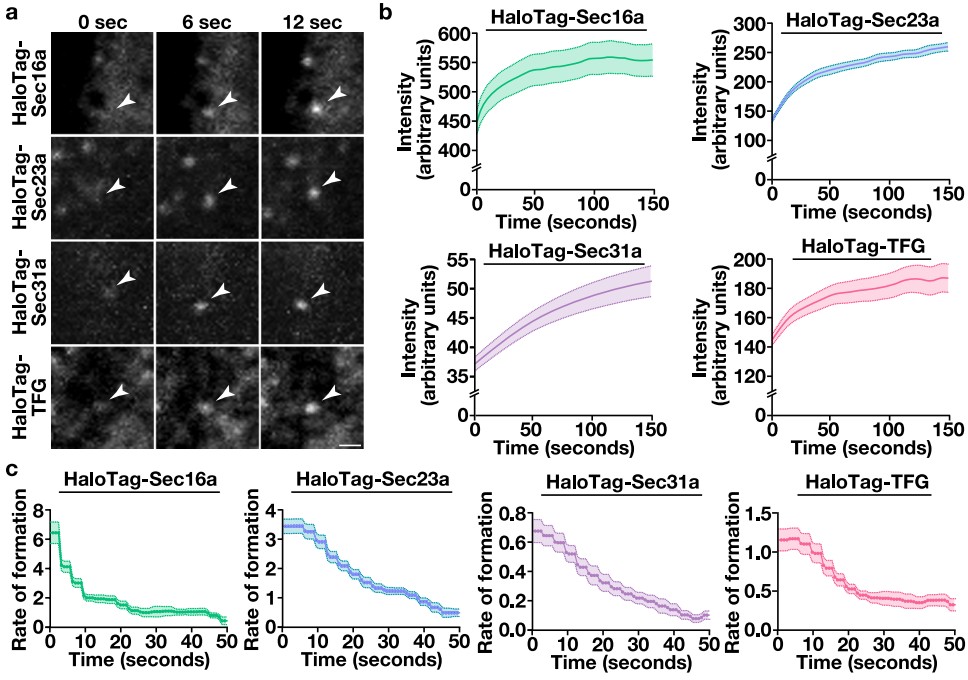

**Fig. 3 | Recruitment kinetics of COPII components and associated regulatory factors. a** Representative timelapse series imaged using lattice light-sheet microscopy showing the accumulation of HaloTag fusion proteins at ER subdomains following labeling with the JFX650-HaloTag ligand. Arrowheads highlight structures undergoing de novo formation in cells. Bar, 1 µm. Quantification of the intensity (**b**) and instantaneous rate of formation (**c**) of newly generated sites decorated by HaloTag fusion proteins (HaloTag-Sec16a, green; HaloTag-Sec23a, light blue; HaloTag-Sec31a, purple; HaloTag-TFG, salmon). Error, as displayed by lightly colored bands around smoothed trend lines, represents mean +/- SEM ($n = 10$ cells each; 3 biological replicates; more than 3000 tracked particles each). Source data are provided as a Source Data file.

stable rate of subunit incorporation (Fig. 3c and Supplementary Fig. 6e). Moreover, Sec23a- and Sec31a-decorated sites exhibited highly distinct rates of formation. In particular, Sec31a subunit addition was approximately 5-fold slower than that of the inner coat, although its incorporation appeared to occur over a longer period of time (Fig. 3c and Supplementary Fig. 6e). Together, these data show that each regulator of COPII-mediated trafficking has a distinctive lifetime, frequency of formation, and rate of incorporation, and that these kinetics can be defined using high spatiotemporal imaging.

## Nutrient availability influences the kinetics of COPII subunit recruitment

Previous studies have shown that acute nutrient limitation reduces the rate of secretory cargo trafficking from the ER, with some ascribing this effect to changes in the expression levels of key COPII regulatory factors[51] and/or their post-translational modification[58,59]. Using ss-DsRed as a model secretory cargo, we confirmed that RPE1 cells deprived of nutrients for two hours exhibited an approximately 2-fold delay in transport to the perinuclear Golgi (Fig. 4a, b and Supplementary Movies 5 and 6), which was identified based on the localization of Golgi matrix protein 130 (GM130) (Supplementary Fig. 7a). Surprisingly however, cells that underwent prolonged nutrient deprivation failed to show a similar effect on the trafficking of ss-DsRed. Instead, the rate of ss-DsRed transport to the Golgi grown in the absence of nutrients for 24 h was comparable to that found in control cells maintained under nutrient replete conditions (Fig. 4a, b and Supplementary Movies 5 and 7). To determine whether this difference in trafficking during acute and prolonged nutrient limitation was universal, we examined another model cargo (ManII-SBP-GFP). In this case, similar trafficking delays were found irrespective of the length of nutrient deprivation (Supplementary Fig. 7a, b).

In light of these opposing findings, we developed an alternative approach to examine ER export under differing nutrient conditions that avoids the use of artificial cargoes and instead focuses on endogenous ERGIC-53, a well-characterized COPII cargo that guides several glycoproteins during their anterograde transport[37,38]. For these studies, we used CRISPR/Cas9-edited cells expressing SNAP-tag-ERGIC-53 from its native locus[67] and following dye-labeling, conducted highly inclined thin illumination (HILO) imaging to monitor changes in its fluorescence intensity at newly formed peripheral sites. We found that SNAP-tag-ERGIC-53 accumulated approximately 60% more slowly following acute nutrient deprivation, as compared to either nutrient replete conditions or when nutrients were absent for a prolonged period (Fig. 4c, d), consistent with our findings examining ss-DsRed trafficking to the Golgi.

One notable difference between control and nutrient deprived cells was the morphology of cargoes that accumulated at the Golgi, which exhibited a fragmented appearance (Fig. 4a). To determine whether this phenotype could be attributed to a change in Golgi morphology under these conditions, we used CRISPR/Cas9 editing to append HaloTag onto endogenous GRASP65, an interactor of GM130 that helps to maintain lateral Golgi ribbon connectivity[73], and performed super-resolution imaging in the presence and absence of nutrients. Our findings revealed that the highly interconnected, ribbon-like morphology of the Golgi was dramatically altered following both 2 and 24 h of nutrient deprivation (Supplementary Fig. 7c), clarifying why the morphology of cargoes near the perinuclear region was perturbed under nutrient limiting conditions.

Our discovery that extending the duration of nutrient deprivation leads to remediation of a cargo trafficking deficit is inconsistent with current models. Previous work showed that glucose starvation leads to downregulation of XBP1 splicing, which normally promotes the expression of several factors involved in COPII-mediated trafficking, including Sec16a and multiple COPII coat subunits[58]. To determine whether prolonged nutrient deprivation may alter levels of spliced XBP1 (sXBP1), we conducted quantitative PCR studies. However, we

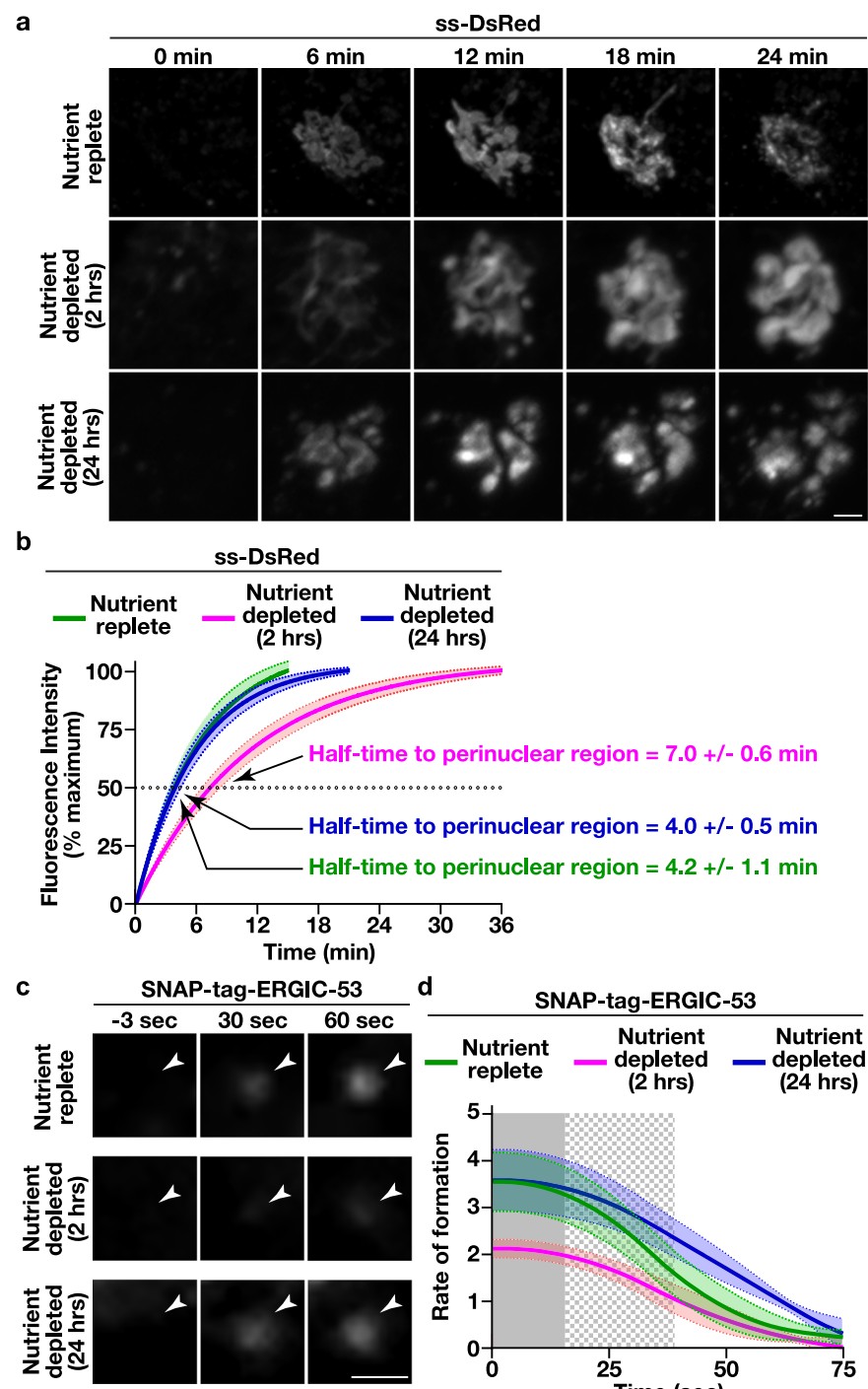

**Fig. 4 | Nutrient deprivation influences the rate of cargo egress from the ER.**
**a** Control cells expressing ss-DsRed were maintained under nutrient replete conditions or subjected to acute (2 h) or long-term (24 h) nutrient deprivation prior to SLF-mediated cargo release and live cell confocal imaging. Representative images zoomed to the perinuclear region of cells are shown at various timepoints. Bar, 1 μm. **b** Confocal imaging of control cells expressing ss-DsRed under various conditions (nutrient replete, green; 2 h nutrient depletion, magenta; 24 h nutrient depletion, blue) was used to monitor its synchronous release from the ER. Based on fluorescence intensity, the percentage of ss-DsRed present within the perinuclear region relative to its maximal accumulation there was quantified over time and fit to an exponential plateau (discontinued upon reaching maximal intensity). Error, as displayed by lightly colored bands around fitted exponential plateau trend lines, represents mean +/- SEM ($n = 20$ cells each; 3 biological replicates each). Half-times to perinuclear accumulation of ss-DsRed were calculated based on the plots (see Source Data file). **c** Cells natively expressing SNAP-tag-ERGIC-53 were maintained under nutrient replete conditions or subjected to nutrient depletion and visualized using HILO imaging. Representative images zoomed to newly forming sites (highlighted by arrowheads) are shown at various timepoints ($n = 20$ cells each; 3 biological replicates each). Bar, 1 μm. **d** The intensities of new sites harboring SNAP-tag-ERGIC-53 were tracked under various nutrient availability conditions (nutrient replete, green; 2 hour nutrient depletion, magenta; 24 h nutrient depletion, blue). Error, as highlighted by lightly colored bands around smoothed trend lines, represents mean +/- SEM (>6000 tracked particles, each condition). Solid gray ($p < 0.05$) and checkered gray ($p < 0.1$) regions represent significant differences, calculated using multiple, unpaired $t$ tests at each time point. Specific $p$ values are provided in the Source Data file.

found no significant change in sXBP1 mRNA levels in cells lacking nutrients for 2 or 24 h as compared to controls (Supplementary Fig. 8a). Consistent with this finding, expression of another effector of sXBP1, the ER chaperone GRP78/BiP, was unchanged between 2 and 24 h of nutrient deprivation (Supplementary Fig. 8b). We next directly examined expression of COPII in the presence and absence of nutrients, using immunoblotting analysis and quantitative live cell imaging. Consistent with prior work[58], the total cellular levels of both inner and outer COPII subunits were downregulated following 2 h of nutrient deprivation and, in the case of the inner coat, were reduced further following 24 h under these conditions (Supplementary Fig. 8c–e). In contrast, TFG levels were only mildly affected by a short-term decrease in nutrient availability, but they were dramatically reduced after prolonged absence of nutrients (Supplementary Fig. 8c–e). Together, these data argue against the idea that the overall expression levels COPII coat components or their regulators directly controls the rate of secretory cargo export.

To determine whether short- and/or long-term nutrient deprivation may have specific impacts on COPII at individual ER subdomains, we conducted a series of quantitative, live-cell imaging studies using confocal microscopy. We found that the total number of sites decorated with COPII subunits was reduced following prolonged nutrient deprivation, but their intensities were significantly elevated as compared to control cells and cells subjected to a short-term loss of nutrients (Supplementary Fig. 9a–d). Similarly, the levels of Sec16a and TFG at ER subdomains were significantly elevated with long-term nutrient deprivation (Supplementary Fig. 9a–d). These data raise the possibility that the local concentration of COPII and/or its regulators at ER subdomains may serve as a key determinant that governs the rate of cargo export.

To investigate this idea, we measured the recruitment kinetics of each HaloTag fusion protein under differing nutrient availability conditions. Following two hours of nutrient deprivation, the lifetime of Sec16a at ER subdomains decreased significantly (approximately 2-fold) as compared to control cells, although its frequency of formation increased (approximately 4-fold), suggesting a high rate of turnover under this condition (Fig. 5a, b). In contrast, after 24 h of nutrient deprivation, Sec16a-labeled sites became hyperstabilized, with a significantly extended lifespan as compared to control cells (approximately 2-fold) (Fig. 5a). Similarly, the lifetime of Sec23a-positive structures increased modestly at the 24 h timepoint, although their frequency of formation declined by nearly 3-fold (Fig. 5a, b). Moreover, the outer COPII coat exhibited both a diminished lifetime and frequency of formation following 24 h of nutrient deprivation, despite the rate of cargo trafficking returning to that of nutrient replete conditions (Fig. 5a, b).

We next conducted an analysis of individual assembly curves in the presence and absence of nutrients. These comparisons revealed that the rate of Sec16a subunit incorporation at ER subdomains was generally elevated following both short- and long-term nutrient deprivation as compared to control cells (Fig. 5c). In contrast, the rate of TFG addition was significantly higher during short-term nutrient elimination as compared to control cells or cells subjected to a long-term absence of nutrients (Fig. 5d). Perhaps most strikingly, Sec23a subunit incorporation at ER subdomains was dramatically enhanced (approximately 1.8-fold) following 24 h of nutrient deprivation, as compared to control cells or cells depleted of nutrients for 2 h (Fig. 5e), with Sec31a exhibiting a similar, albeit more modest, trend (Fig. 5f). These data suggest that the rate of Sec23a subunit addition at ER subdomains may serve as a key control point in determining the efficiency of anterograde transport of newly synthesized secretory cargoes.

**Artificially increasing the rate of Sec23a subunit incorporation bypasses cargo trafficking deficits that occur following short-term nutrient deprivation**

To directly test the idea that the kinetics of Sec23a subunit addition at ER subdomains functions as a rheostat that determines the speed of secretory cargo efflux, we required an approach to artificially increase its rate of incorporation. Based on previous work, we found that overexpression can serve as a simple but effective means to alter how quickly a factor becomes concentrated at a membrane surface[74]. We therefore stably transduced cells natively expressing HaloTag-Sec23a with a virus encoding GFP-Sec23b. While fluorescence microscopy demonstrated that overexpressed GFP-Sec23b was incorporated into sites harboring endogenous HaloTag-Sec23a (Fig. 6a), live cell imaging failed to reveal significant changes in HaloTag-Sec23a lifetime, its frequency of formation, or its overall level of expression (Supplementary Fig. 10a–c). Nonetheless, the rate of HaloTag-Sec23a subunit addition was elevated significantly by overexpression of GFP-Sec23b, to a level similar to that observed following 24 h of nutrient deprivation (Fig. 6b, c). Strikingly, the delay in ss-DsRed trafficking associated with short-term nutrient deprivation was resolved by solely overexpressing GFP-Sec23b (Fig. 6d and Supplementary Movie 8). To further validate these findings, we measured de novo SNAP-tag-ERGIC-53 accumulation at peripheral ER subdomains following GFP-Sec23b overexpression in cells acutely depleted of nutrients. Under these conditions, SNAP-tag-ERGIC-53 intensity was increased by approximately 1.5-fold relative to mock-treated control cells (Supplementary Fig. S10d, e). Together, these data strongly suggest that the kinetics of Sec23 incorporation at ER subdomains controls the rate at which cargoes are able to exit the ER.

## Discussion

Similar to gene transcription and protein translation, the nearly constitutive export of newly synthesized secretory proteins from the ER is essential for normal cellular homeostasis, growth, and development[7]. Importantly, a variety of external stimuli have been shown to directly regulate each of these processes, enabling cells to react to changing environmental conditions[51–65]. For example, in response to an immune challenge, B cells undergo differentiation to generate antibody-secreting plasma cells, which requires a dramatic change in gene expression patterns, expansion of the organelles that participate in membrane protein trafficking, and an increased flux through the COPII-mediated early secretory pathway[75]. While several signaling networks that drive proliferation of ER membranes and/or modulate COPII subunit expression have been defined, mechanisms that directly control the rate of secretory protein packaging and export at individual sites of COPII transport intermediate biogenesis have remained poorly understood. To address this challenge, previous studies have relied mainly on the analysis of fixed cells grown under various conditions to describe impacts to the formation of ER subdomains capable of secretory protein trafficking. In particular, the steady state levels of COPII components at these sites and the total number of subdomains found in cells have been used as a proxy for determining biosynthetic cargo export activity[51,59]. While this work has been invaluable to identify regulators of COPII-mediated transport, the lifetime of COPII budding sites, their frequency of formation, and their rate of subunit recruitment have not been defined. Here, we reveal the dynamics of the COPII coat complexes under differing environmental conditions, showing that Sec23, which regulates GTP hydrolysis on Sar1, specifically plays a key role in coordinating the timing of transport intermediate biogenesis, even when the overall levels of other COPII components and regulators are reduced.

Under nutrient limiting conditions, several studies have identified key changes in Sec23 post-translational modification, which influence its ability to associate with other factors involved in COPII-mediated trafficking[52,53,56,62]. In particular, phosphorylation of Sec23b by ULK1 under conditions identical to those used in our nutrient deprivation studies, leads to its redistribution to ERGIC membranes, where it functions in the biogenesis of autophagosomes[53]. Similarly, other forms of post-translational modification have been suggested to alter the ability of Sec23 to target to ER subdomains that produce COPII

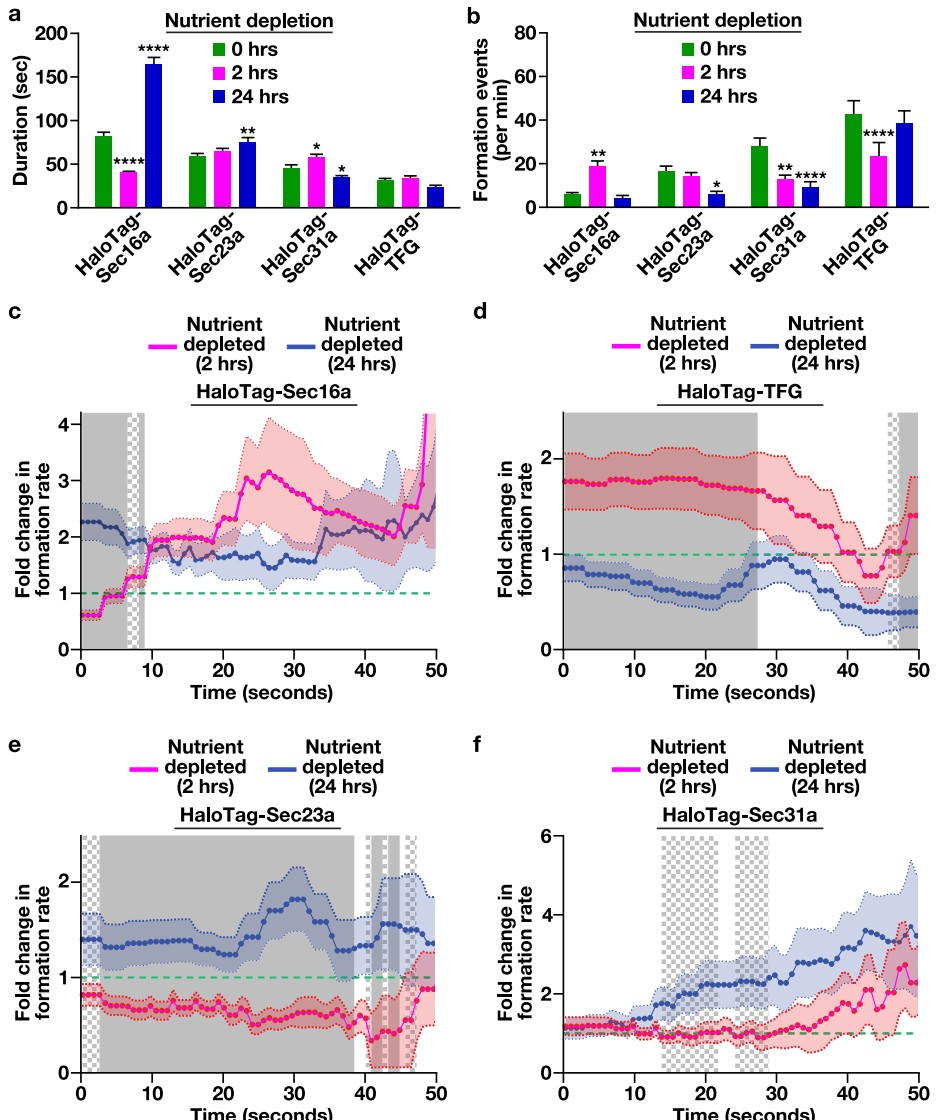

**Fig. 5 | Short-term nutrient deprivation slows inner COPII coat recruitment to ER subdomains.** Quantification of the average duration of each HaloTag fusion protein at an ER subdomain (imaged using lattice light-sheet microscopy) under various nutrient availability conditions (**a**) or the number of new structures that form each minute under those conditions (**b**). As indicated in the figure panels: nutrient replete, green; 2 h nutrient depletion, magenta; 24 h nutrient depletion, blue. Error bars represent mean +/- SEM ($n = 12$ cells with more than 10,000 tracked particles each; 3 biological replicates). ****$p < 0.0001$, **$p < 0.005$, and *$p < 0.05$, as calculated using a one-way ANOVA and Tukey post hoc test. Specific $p$ values are provided in the Source Data file. **c–f** Quantification of the fold change in HaloTag fusion protein assembly rates following 2 and 24 h nutrient deprivation (2 h nutrient depletion, magenta; 24 h nutrient depletion, blue) relative to nutrient replete conditions (represented by a dashed green line). Error, as displayed by lightly colored bands around smoothed tread lines, represents mean +/- SEM ($n = 10$ cells each; 3 biological replicates; more than 10,000 tracked particles each). Solid gray ($p < 0.05$) and checkered gray ($p < 0.1$) regions represent significant differences, calculated using multiple, unpaired $t$ tests at each time point. Specific $p$ values are provided in the Source Data file.

transport intermediates[52,56,62]. Although relatively few studies have investigated impacts of prolonged nutrient deprivation, there is clear precedent for cells to adapt to environmental conditions, which may ultimately lead to further modifications on Sec23 that enable it to more efficiently support transport intermediate formation, even when its overall levels are dramatically lower[76]. Further studies will be necessary to define the importance of potential sites of post-translational modification on Sec23 under varying nutrient availability conditions to validate this idea. Additionally, defining Sec24 dynamics would be of interest, although they may vary depending on the paralog and differ from Sec23 due to their roles in cargo-mediated retention.

Based on our live cell imaging studies, the inner COPII coat persists at ER subdomains for nearly 60 seconds on average, similar to the 45–60 s necessary for clathrin-mediated endocytic pits to form and internalize from the surface of cells[77]. In contrast, in vitro

reconstitution studies suggest that the lifetime of assembled Sar1-Sec23-Sec24 complexes on membranes is substantially more transient, on average approximately 30 s[36]. While we cannot exclude the possibility that the sites we identified in cells consist of multiple, simultaneous budding events, this is unlikely given their narrow range of individual lifetimes. Instead, the reconstituted inner COPII coat may disassemble prematurely in vitro due to the absence of regulatory factors, which influence COPII dynamics and GTP hydrolysis on Sar1. Our studies also revealed the relative longevity of some of these regulators at ER subdomains, indicating that Sec16a persists for a longer period of time as compared to COPII. These data support a model in which Sec16a functions to establish sites of COPII intermediate budding and facilitates the sequential recruitment of inner and outer COPII coat complexes. Consistent with this idea, Sec23 persists for a longer period of time at ER subdomains as compared to Sec31. Notably, the

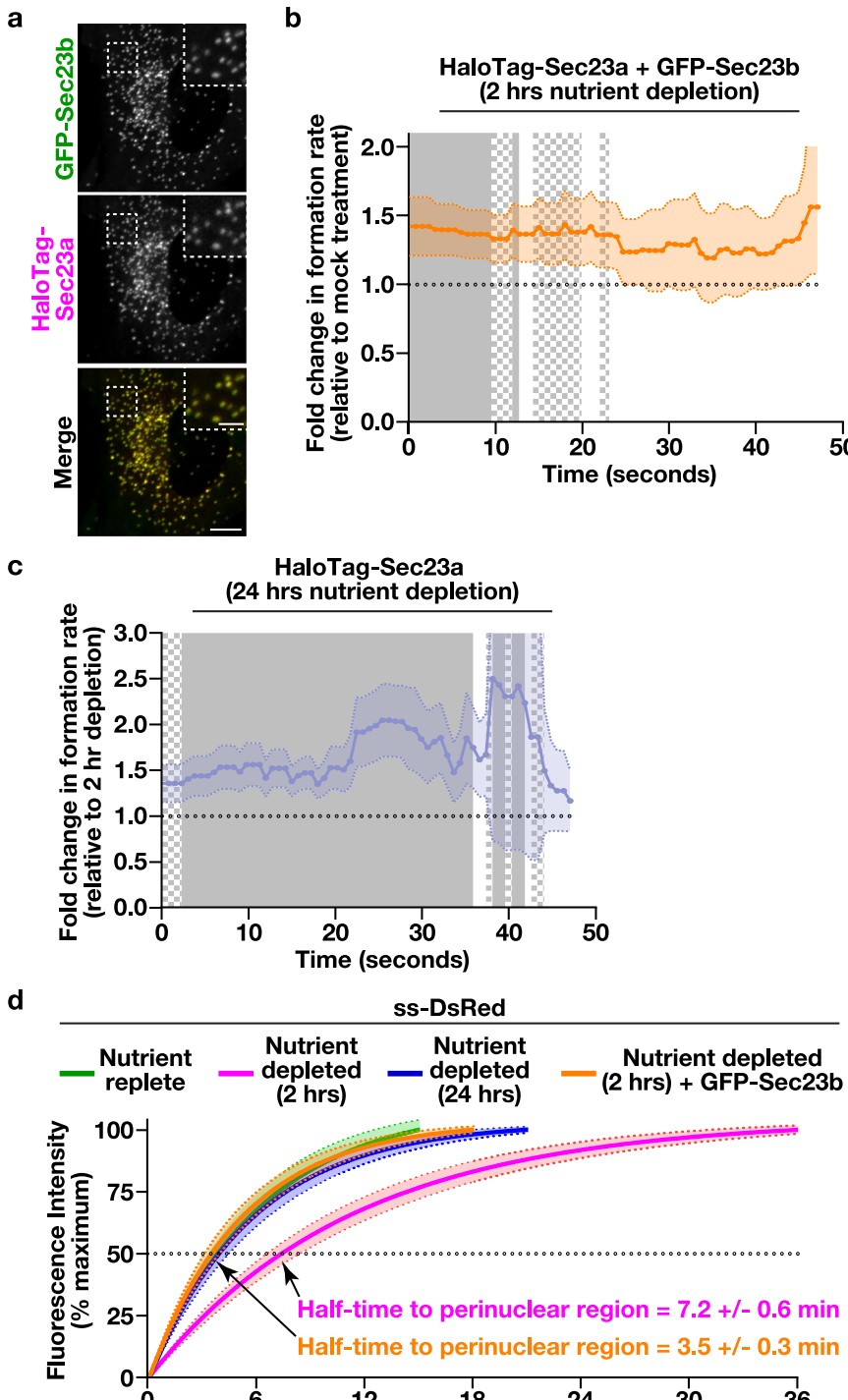

**Fig. 6 | Increasing the rate of inner COPII coat recruitment accelerates cargo egress from the ER during acute nutrient deprivation. a** Representative confocal images showing the relative distribution of GFP-Sec23b (green) as compared to natively expressed HaloTag-Sec23a following labeling with JFX650-HaloTag ligand (magenta). Bar, 5 μm; inset bar, 1 μm. Quantification of the fold change in the formation rate of structures harboring HaloTag-Sec23a following 2 h nutrient deprivation in the presence (orange) and absence of exogenously expressed GFP-Sec23b (**b**) or 24 h nutrient depletion (blue), as compared to the 2 h nutrient depletion condition (**c**). Error, as displayed by lightly colored bands around smoothed trend lines, represents mean +/- SEM (*n* = 10 cells each; 3 biological replicates; more than 9000 tracked particles each). Solid gray (*p* < 0.05) and checkered gray (*p* < 0.1) regions represent significant differences, calculated using multiple, unpaired *t* tests at each time point. Specific *p* values are provided

in the Source Data file. **d** Confocal imaging of cells expressing ss-DsRed under various conditions (nutrient replete, green; 2 h nutrient depletion, magenta; 24 h nutrient depletion, blue), including the presence of exogenously expressed GFP-Sec23b (orange), was used to monitor its synchronous release from the ER, as described in panel 4B. Based on fluorescence intensity, the percentage of ss-DsRed present within the perinuclear region relative to its maximal accumulation there was quantified over time and fit to an exponential plateau (discontinued upon reaching maximal intensity). Error, as displayed by lightly colored bands around fitted exponential plateau trend lines, represents mean +/- SEM (*n* = 20 cells each; 3 biological replicates each). Half-times to perinuclear accumulation of ss-DsRed following 2 h of nutrient deprivation were calculated in the presence and absence of exogenously expressed GFP-Sec23b (see Source Data file).

longest-lived components exhibited the fewest formation events, while those that appeared relatively transiently arose more often in cells, indicative of an inverse relationship between lifetime and frequency of formation. This balance may help to ensure that at any given time, each component of the COPII machinery is found at nearly every ER budding site, which is commonly witnessed at steady state, despite the highly dynamic nature of COPII assembly.

To our surprise, the recruitment kinetics of the COPII regulatory factor TFG were very distinct from its binding partner Sec23, exhibiting a significantly shorter lifetime at the ER/ERGIC interface. In previous work, we demonstrated that TFG functions to cluster COPII transport intermediates, while promoting their uncoating prior to fusion with acceptor membranes[44,45]. Following short-term nutrient deprivation, when COPII-mediated trafficking was slowed, the rate of TFG recruitment was paradoxically elevated, suggesting that TFG function may extend beyond its canonical roles in facilitating transport intermediate fusion with ERGIC membranes. One possibility is that increased TFG addition under nutrient limiting conditions interferes with COPII outer coat assembly, as TFG competes with Sec31 for binding to Sec23. This idea is consistent with the decreased frequency of Sec31 appearance following acute nutrient depletion. Whether TFG may function similarly under nutrient replete conditions to regulate COPII coat assembly will require additional study. Nevertheless, based on our work, we now have ideal platforms to resolve this and other questions focused on the regulation of COPII dynamics.

Although not a major focus of this study, we also identified a large number of highly transient structures decorated by COPII subunits, which were visible for less than 10 s. These sites may represent failed nucleation events, similar to those witnessed during clathrin-mediated endocytosis (CME)[78,79], raising the possibility that regulatory mechanisms exist to promote COPII bud maturation. However, in contrast to CME, it appears that stable ER subdomains exist[80], based on the presence of numerous long-lived sites that persist well beyond the imaging window used in our study (ie., 10 min). In the absence of sub-diffraction imaging, it remains unclear whether these domains represent one or more budding events, but with the advent of more sensitive, rapid, and high resolution microscopy approaches, it should soon become possible to define precisely how COPII can manage the plethora of cargoes that must be exported continuously from the ER.

Also unexpected was our finding that the two artificial cargoes we analyzed (ss-DsRed and MBP-SBP-GFP) traffic distinctly following prolonged nutrient deprivation. This difference may be due to unique impacts on their individual cargo receptors[1–4,81,82]. Alternatively, the biotin-based, synchronized release system may be more dependent on nutrient availability as compared to SLF-mediated export, resulting in an indirect delay in transport under the conditions we used. Now with the ability to study a native COPII client (SNAP-tag-ERGIC-53), we can avoid the use of artificial cargo release systems, enabling a more physiological view of trafficking in the early secretory pathway. In the future, it will be critical to expand the repertoire of cargoes that can be examined in mammalian cells, without a need for overexpression or aggregation in the ER lumen prior to release.

## Methods

### Cell culture and genome editing

CRISPR/Cas9-mediated genome editing of human hTERT-immortalized RPE1 cells (CRL-4000 from ATCC) grown in DMEM:F-12 media (nutrient replete) was conducted as described previously[67,74]. Briefly, cells were transfected with purified Cas9, a specific gRNA, and a homology directed repair (HDR) template. The following guide RNA (gRNA) sequences were used: 5′-GACGGGACCGTCTGGGGCGG-3′ (targeting Sec16a), 5′-CTTTAACTTCATCCTGCTAA-3′ (targeting Sec31a), 5′- ATCCAACTGTCCGTTCATGG-3′ (targeting TFG), and 5′-TCTACCACAGAATAACACCC-3′ (targeting GRASP65). Other edited cell lines have been described elsewhere[67]. Clonal populations were

isolated using fluorescence activated cell sorting (FACS), which was followed by live cell imaging to confirm the proper distribution of each fusion protein. Cell line authentication is carried out using Short tandem repeat (STR) analysis (annually), demonstrating that all lines used were RPE1 cells.

To measure proliferation rates, cells seeded at low density were treated with 2 μM Calcein AM (15 min every 24 h) and imaged on an ImageXpress Micro 4 platform (Molecular Devices) using a 10x (0.3 NA) dry objective. MetaXpress software was used to count cells in an unbiased manner, and proliferation rates were determined using GraphPad Prism software. To conduct nutrient deprivation studies, cells were washed and subsequently cultured in Earle's Balanced Salt Solution (EBSS) for 2 or 24 h.

### Fluorescence microscopy and image analysis

Confocal and HILO imaging studies were carried out on a Nikon ECLIPSE Ti2 spinning disk confocal microscope equipped with a 60x oil immersion objective (1.4 NA), a Hamamatsu ORCA-Flash4.0 sCMOS camera, and an iLas2 total internal reflection fluorescence (TIRF) imaging system with photostimulation capabilities. Nikon Elements software was used for image acquisition. HaloTag fusion proteins were labeled using either JFX650- or JF635-HaloTag ligands, and SNAP-tag fusion proteins were labeled using JFX650-SNAP-tag ligand (generously provided by Dr. Luke Lavis). Cargo trafficking and immunostaining assays were performed as described previously[67]. Briefly, cargo release was mediated by the addition of 50 μM SLF or biotin (200 nM), as indicated in the Results section. Validated antibodies (1 μg/mL) directed against Sec24a (Santa Cruz Biotechnology; sc-169279), GM130 (BD Sciences; BD610823), ERGIC-53 (Santa Cruz Biotechnology; sc-66880), and PDIA3 (Proteintech; 15967-1-AP) were used for immunofluorescence studies. All datasets acquired during imaging were Nyquist sampled.

For HILO imaging, an imaging depth of 2 μm was achieved by directing an excitation laser at a sharp angle determined empirically. Photobleaching studies were conducted using a Leica SP8 3X STED system equipped with a 60x oil immersion objective (1.4 NA) and a super-continuum white-light laser. Super resolution imaging was performed on a Zeiss LSM 880 confocal system with Airyscan using a 63x (1.4 NA) oil immersion objective. Zeiss ZEN software was used for image acquisition. Imaging datasets were acquired at Nyquist sampling using ZenBlue Z sampling recommendations, following by deconvolution and denoising using proprietary Zeiss algorithms. Light-sheet microscopy was conducted on a 3i Lattice LightSheet microscope equipped with a Hamamatsu ORCA-Flash4.0 V3 sCMOS camera. Beam alignment, dye alignment, and bead alignment were performed prior to imaging. All timelapse light-sheet studies were conducted with temporal resolutions of 3–4 s, 1024 × 1024 pixel resolution, and a step size of 297 nm in the Z plane. Images were deskewed and deconvolved using Slidebook software (3D frequency filter enabled, Gaussian smoothing at 0.6, and mirrored edge padding of 20%).

Fiji image processing software was used to quantify the accumulation of fluorescent cargoes within the perinuclear region of cells (relative to their fluorescence maxima measured during imaging) and conduct linescan measurements. Cargo accumulation over time was fitted to an exponential plateau equation (GraphPad Prism). Similarly, Pearson's correlation analysis was also carried out using GraphPad Prism. Imaris software was used to measure intensities, volumes, and the density of structures within cells. To quantify the dynamics of HaloTag fusion proteins at ER subdomains, lattice light-sheet imaging datasets were time normalized, and only structures that assembled and disassembled during each imaging session were analyzed. GraphPad Prism was used to generate frequency histograms, which were fitted to a two-phase exponential decay equation. Assembly curves were generated based on intensity measurements and smoothed using fine Locally Weighted Scatterplot Smoothing

(LOWESS) and fitted to exponential plateau equations to calculate rate constants (GraphPad Prism). Derivatives were calculated using GraphPad Prism and used for statistical analysis. Unless otherwise noted, images shown are representative of 3 biological replicates (minimally 10 cells each).

### Biochemistry and molecular biology

Immunoblotting studies were conducted as described previously[45]. Briefly, samples were separated by SDS-PAGE, followed by transfer to nitrocellulose and proteins were detected using SuperSignal West Femto Maximum Sensitivity Substrate (Thermo Scientific; 34094). The following validated antibodies were used: Sec31a (BD Sciences; 612351), Sec23a (Thermo Scientific; PA5-28984), TFG (Novus Biologicals; NBP2-62212), β-actin (Sigma; A1978), GAPDH (Proteintech; 60004-1), and GRP78/BIP (Proteintech; 11587-1-AP).

RNA extraction was performed using TRIzol (Invitrogen), followed by sequential precipitations using ethanol and lithium chloride. Production of cDNA was performed using a Superscript III First Strand RT-PCR kit (Invitrogen), and RT-qPCR experiments were performed using a CFX384 Touch Real-time PCR detection system (Bio-Rad) and Applied Biosystems Power SYBR Green PCR Master Mix (Thermo Scientific). The following primers were used: sXBP1 F: CCCTCCAGAACATCTCCCCAT; sXBP1 R: ACATGACTGGGTCCAAGTTGT; GAPDH F: AGCCACATCGCTCAGACAC; GAPDH R: GCCCAATACGACCAAATCC.

### Statistics and reproducibility

All statistical tested used are highlighted in figure legends. No statistical method was used to predetermine sample size. Instead, sample sizes were determined based on previous imaging studies[74]. No data were excluded from the analyses, and all attempts at replication were successful. Experiments conducted were randomized, and the investigators were blinded to allocation during experiments and outcome assessment.

### Reporting summary

Further information on research design is available in the Nature Portfolio Reporting Summary linked to this article.

## Data availability

All reagents generated in this study are available from the corresponding author with a Material Transfer Agreement. All data supporting the findings of this study are available within the article and Supplementary Information files. Source data are also provided with this paper. Source data are provided with this paper.

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

## Acknowledgements

This work was supported in part by NIH Grant GM134865 (to A.A.). P.L. and K.A.S. received support from T32 GM008688. Cell sorting studies were conducted in the UWCCC Flow Cytometry Laboratory (supported by 1S10RR025483-01), and some imaging studies were conducted using instrumentation in the UW Optical Imaging Core. Initial lattice light-sheet imaging studies was conducted at the Advanced Imaging Center at the Janelia/Howard Hughes Medical Institute (HHMI), which is a jointly funded venture of the Gordon and Betty Moore Foundation and the HHMI. We thank members of the Audhya lab for critically reading this manuscript.

## Author contributions

W.K., P.L. and A.A. designed research; W.K., P.L. and K.A.S. performed research; W.K., P.L. and A.A. contributed reagents/analytic tools; W.K., P.L. and A.A. analyzed data; and W.K. and A.A. wrote the paper.

## Competing interests

The authors declare no competing interests.
