## [Peer Review File · Nature Communications]

Nutrient deprivation alters the rate of COPII subunit recruitment at ER subdomains to tune secretory protein transportREVIEWER COMMENTS

Reviewer #1 (Remarks to the Author):

The authors, Kasberg et al., utilized CRISPR/cas9 to generate endogenously expressed HaloTag-COPII subunits (Sec16a, Sec23a, Sec31a, TFG), and investigated the spatiotemporal dynamics of these subunits under different nutrient conditions. By leveraging these tools, the authors found that the subcellular local concentration of COPII subunits and/or its regulator governs the rate of cargo export, rather than the overall expression level of COPII subunits. Towards the end of the paper, the authors transduced an exogenously overexpressed GFP-Sec23b or GFP-Sec23b R722A into a Halo-Tag-Sec23a cell line, aiming to demonstrate that the Sec23 incorporation at ER subdomains controls the rate of ER cargo export. However, the Sec23b R722A GAP activity loss-of-function mutant appears disconnected and confuses the overall story. Overall, this study provides valuable insights into COPII coat complex dynamics under varying conditions and highlights that the local concentration of COPII subunits controls the rate of ER cargo transport, providing new insights to the field. Nevertheless, in some parts, the logic is untenable, and several pieces of evidence need further strengthening.

1, It is intriguing that in different HaloTag-edited cell lines, there are consistently long-lived and short-lived sites. What distinguishes these two types of sites, and is there a standard for differentiating between them?

2, Considering the vague description of long-lived and short-lived sites, I am curious if these sites vary linearly, which means the duration of the site you select could be random. The authors stated that "To simplify analysis...", but I have concerns about the representativeness of the images, as the selected window usually only contains a single site that may differ from the majority of signals in the cell. A more convincing approach would be to conduct a comprehensive analysis of a single cell's signals and to repeat the analysis on enough samples, rather than randomly selecting a "window." A precise biological process need be quantified more carefully.

3, Figure 1 indicates that the sec16a-halotag system may pose problems and interfere with the experiment. Despite this, the cell line is still utilized in subsequent experiments and to explain certain phenomena. This significantly undermines the reliability of the conclusions drawn from such experiments.

4, In Figure 4, the authors found that cargo transport to the perinuclear region was comparable after 24 hours of nutrient deprivation and under nutrient replete conditions. However, this conclusion was drawn based on the ss-DsRed indicator, rather than the ManII-SBP-GFP indicator shown in Supplementary Figure 5b and 5c. Additionally, Supplementary Figure 5e and 5f demonstrate a similar pattern. Obviously, ss-DsRed signals is totally different with ManII-SBP-GFP signals at 24h deprivation condition. What do these changes signify, and is there any physiological explanation? If not the Golgi, what do the perinuclear regions indicated represent, and where are the cargos transported to?

5, At the end of the paper, the authors attempted to demonstrate that a gain-of-function in Sec23a assembly rate could overcome cargo trafficking deficits. However, I did not see any evidence of artificially

increased Sec23a assembly rate in Supplementary Figure 8a-b, where the duration and assembly events of Sec23a were unchanged. Moreover, in Figure 6e, the authors showed that GFP-Sec23b R722A had a deficit in cargo transport compared to WT-Sec23b, which left me puzzled about the logic behind the claim. It would be more precise if the authors included a GAP activity gain-of-function mutant and a vector control in their experiments. Additionally, while the authors used ss-DsRed as a cargo indicator in Figure 6e, what about ManII? Only when the cargos move normally from the ER to the Golgi can it be considered "bypassing cargo trafficking deficits." Or it could be somehow an artificial phenomenon.

6, The author underscores the potential bias of the expression system at the outset, opting for the endogenous system. However, in Figure 6, the author resorts to the overexpression system to clarify some phenomena, which presents a degree of self-contradiction.

Minor points:

1, The description of Figures 4b and 4c appears to be missing in the main text.

2, The authors first mentioned the Sec23b R722A mutant as a GAP activity loss-of-function mutant, stating: "To determine whether the GAP activity of Sec23 is necessary for this effect, we overexpressed a mutant form of GFP-Sec23b, which harbors the R722A mutation that lacks GAP activity in vitro (49)." They also cited a reference (49) by Zacharogianni et al. in EMBO Journal 2011. However, I could not find any description of Sec23b GAP activity or the R722A mutant in the paper. Are the authors trying to emphasize a different point here?

Reviewer #2 (Remarks to the Author):

In this paper, Kasberg et al. examined dynamics of COPII assembly and secretory cargo export from the ER and the effects of nutrient deprivation on these processes. By live imaging, they found that the rate of inner COPII coat assembly is important for determining the pace of cargo export rate. They also showed that the increase of inner coat assembly kinetics is sufficient to rescue cargo export deficiency caused by acute nutrient limitation.

This is a very interesting piece of work. I am impressed by how carefully the authors designed the experimental system to avoid possible artifacts caused by expression of tagged proteins. The expression of HaloTag-Sec16a appeared to be a little hazardous but the authors honestly described it. The effects of nutrient starvation are very complicated, but the authors' interpretations appear to be reasonable, although I am not 100% convinced that they are the only possible explanations.

The results of this kind of detailed analysis on COPII assembly in vivo should be valuable for researchers in this field and therefore I am basically supportive for the publication of this work. However, I would like

to ask the authors to address the following comments and questions before the final acceptance for publication.

1. Trafficking of ss-DsRed is compromised by 2-h nutrient depletion but recovers after 24 h. However, that of MnII-SBP-GFP appears to be more defective at 24 h. On the other hand, L1CAM shows even higher efficiency of transport after 24-h nutrient depletion. How do the authors explain these differences?

2. The authors observed GRASP65 to reveal the Golgi morphology, saying that it is involved in cisternal stacking. However, there are arguments on the roles of GRASP proteins and the authors should be careful in the interpretation of the results.

3. The authors argue the effect of GFP-Sec23b by comparing two different figures (e.g. Fig. 5e and 6b, Fig. 3b and 6c). This is not fair. Readers cannot evaluate it objectively.

Reviewer #3 (Remarks to the Author):

The manuscript "Nutrient deprivation alters the rate of COPII coat assembly to tune secretory protein transport" by Kasberg et al. describes changes in several COPII and COPII-related dynamics resulting from nutrient deprivation that culminates in affecting transport rates. I apologize in advance for sounding harsh. In my humble opinion, this manuscript suffers from several problematic issues, both technical and scientific. I will address a few technical issues, although they usually belong to the "minor comments" part. Most of the references are provided in clusters making a follow-up almost impossible. Also, it almost seems that this is a means for the authors to avoid potentially controversial issues, and in so doing, they missed vital manuscripts that would have helped in describing at least some of the data presented. I will be more specific later in this review.

All figures are not numbered, making the review process annoying. Also, the figure legend lacks information to the extent that while writing this review, in some cases, I did not understand what the experiments performed based on the figures. In some cases, the authors send the reader to a reference w/o even briefly describing the experimental details. Moreover, while using many types of state-of-the-art equipment, the authors use the term quantitative while very few numbers are presented. Many graphs can be fitted to various simple forms of exponential equations, and the rate constant (Sec⁻¹) or inverse (Sec) may provide hard numbers for comparing the effect of the various treatments.

A significant issue is the term "assembly," used numerous times throughout this manuscript. Unless I did not understand the data presented, the authors seem not to understand that COPII components

dynamically bind to membranes meaning that the accurate description provided should be on/off rates. I also fail to understand the data as, from this reviewer's experience, the "ER subdomains" known for many decades as transitional ER or better ER exit sites are long-lived at the scale of at least tens of minutes.

Relying only on the RUSH system may pose a problem as regardless of its advantages, it uses an artificial retention mechanism entirely unrelated to the natural ones. They should consider using the thermoreversible folding mutant of Vesicular Stomatitis virus G or collagen.

There are other issues to address, but I will end by directing the authors to two manuscripts they cite buried in clusters, one in Cell (39) and the other in JCB (40), providing a new model for the localization and mechanism of function of COPII. This is also summarized in a recent review 1. In my opinion, this new dogma for how the early secretory works may very elegantly explain the major findings of this manuscript. On a final positive note, the finding that rapid coat on/off dynamics promotes a faster transport rate is potentially very interesting as it may provide mechanistic information on how COPII selects and sorts cargo proteins.

Reference:

1. Malis, Y., Hirschberg, K., and Kaether, C. (2022). Hanging the coat on a collar: Same function but different localization and mechanism for COPII. *Bioessays* 44, e2200064. [10.1002/bies.202200064](https://doi.org/10.1002/bies.202200064).

Response to Reviewers' comments:

Reviewer #1 (Remarks to the Author):

The authors, Kasberg et al., utilized CRISPR/cas9 to generate endogenously expressed HaloTag-COPII subunits (Sec16a, Sec23a, Sec31a, TFG), and investigated the spatiotemporal dynamics of these subunits under different nutrient conditions. By leveraging these tools, the authors found that the subcellular local concentration of COPII subunits and/or its regulator governs the rate of cargo export, rather than the overall expression level of COPII subunits. Towards the end of the paper, the authors transduced an exogenously overexpressed GFP-Sec23b or GFP-Sec23b R722A into a Halo-Tag-Sec23a cell line, aiming to demonstrate that the Sec23 incorporation at ER subdomains controls the rate of ER cargo export. However, the Sec23b R722A GAP activity loss-of-function mutant appears disconnected and confuses the overall story.

We agree with the reviewer and have removed all description of studies involving the Sec23b R722A mutant.

Overall, this study provides valuable insights into COPII coat complex dynamics under varying conditions and highlights that the local concentration of COPII subunits controls the rate of ER cargo transport, providing new insights to the field.

We appreciate this comment, which nicely summarizes a very important aspect of our study.

Nevertheless, in some parts, the logic is untenable, and several pieces of evidence need further strengthening.

Please see responses below to specific points raised.

1, It is intriguing that in different HaloTag-edited cell lines, there are consistently long-lived and short-lived sites. What distinguishes these two types of sites, and is there a standard for differentiating between them?

This is an excellent point. To address this comment, we applied correlation analyses and found that longer lifetimes covaried with the diameter and intensity of sites harboring each HaloTag fusion protein (see new Figure S4). Additionally, we found that sites that exhibited a lifetime, which exceeded our imaging window (10 minutes), also tended to be larger and more intense than those that formed and ultimately dissolved during our imaging window. This raised the concern that these very long-lived sites are actually comprised of multiple ER subdomains that fall below the diffraction limit of light, which have been previously observed using electron microscopy-based studies. As the presence of multiple budding events occurring at a single site would confound our analysis, we opted to focus on short-lived sites, which are more likely to exhibit individual budding events.

2, Considering the vague description of long-lived and short-lived sites, I am curious if these sites vary linearly, which means the duration of the site you select could be random. The authors stated that "To simplify analysis...", but I have concerns about the representativeness of the images, as the selected window usually only contains a single site that may differ from the majority of signals in the cell. A more convincing approach would be to conduct a comprehensive analysis of a single cell's signals and to repeat the analysis on enough samples,

rather than randomly selecting a "window." A precise biological process need be quantified more carefully.

To address this important point, we generated histograms showing the relative frequencies of different site lifetimes. These analyses demonstrated that the distributions of lifetimes for each factor are modeled well by exponential decreases with a combination of slow and fast events (i.e., 2 phase exponential decay). Please see new Figure S5. We calculated the time constants of both slow and fast events (also reported in Figure S5) and found that the slow time constants (i.e., sustained events) are similar to the mean lifetimes of the different HaloTag fusion proteins. We very much appreciate the reviewer raising this point, as our addition quantitative analysis has dramatically improved the quality of the manuscript.

3, Figure 1 indicates that the sec16a-halotag system may pose problems and interfere with the experiment. Despite this, the cell line is still utilized in subsequent experiments and to explain certain phenomena. This significantly undermines the reliability of the conclusions drawn from such experiments.

We agree with the reviewer that the HaloTag-Sec16a fusion protein is likely to be only partially functional. When generating this cell line, which harbors the HaloTag at the amino-terminus of Sec16a, we extensively screened for cell lines that were edited in a homozygous fashion, but we were ultimately unsuccessful. Notably, the carboxyl-terminus of Sec16a has been shown to bind to several COPII subunits, as well as Sec12, and our attempts to add a HaloTag at this site resulted in mislocalization of the fusion to the cytosol. Thus, given the proper localization of the amino-terminally tagged Sec16a, we decided to examine this fusion to enable comparisons to Sec23a, Sec31a, and TFG. Nevertheless, to determine whether the HaloTag impairs the mobility of Sec16a, we carried out photobleaching studies to see whether the fusion we used behaved similarly to those reported in previous studies. Based on this work, we showed that the immobile fraction and half-time to recovery of HaloTag-Sec16a were $38.9 \pm 9.6\%$ and 3.1 ± 0.9 seconds respectively (see new panels in Figure 1). These values are very similar to previous reports, which demonstrated an immobile fraction of $43 \pm 4.5\%$ and half-time to recovery of 5.81 ± 0.36 seconds (DOI: 10.1242/jcs.044032). While not perfectly equivalent to studying untagged Sec16a, which is currently infeasible, we believe our HaloTag-Sec16a fusion expressed below native levels represents the best reagent currently available to study Sec16a dynamics. As Reviewer 2 noted, we have included an explicit description of the drawbacks and potential hazards of data collected using this cell line and leave it up to the reader to determine how much weight to give these results based on the experiments we conducted.

4, In Figure 4, the authors found that cargo transport to the perinuclear region was comparable after 24 hours of nutrient deprivation and under nutrient replete conditions. However, this conclusion was drawn based on the ss-DsRed indicator, rather than the ManII-SBP-GFP indicator shown in Supplementary Figure 5b and 5c. Additionally, Supplementary Figure 5e and 5f demonstrate a similar pattern. Obviously, ss-DsRed signals is totally different with ManII-SBP-GFP signals at 24h deprivation condition. What do these changes signify, and is there any physiological explanation?

The reviewer is absolutely correct to point out that ss-DsRed and ManII-SBP-GFP are trafficked at different rates following prolonged nutrient depletion. In our view, this discrepancy could be explained in several ways, mostly likely of which are: (1) each cargo uses a distinct receptor to exit the ER, and these receptors are differentially impacted by long-term nutrient depletion or (2) differences in the artificial release systems used result in distinct behaviors of the cargoes. Neither of these scenarios is trivial to address. Thus, we leverage a natively expressed cargo

(SNAP-tag-ERGIC-53) to avoid use of artificial release systems that remain relatively poorly characterized. In contrast, ERGIC-53 is a known client of COPII, with crystallography data available to demonstrate how it engaged with the coat complex. Using this fusion protein in combination with highly inclined thin illumination (HILO) imaging, we were able to demonstrate that long-term nutrient deprivation 'rescues' trafficking defects observed following a 2 hour depletion, similar to that seen with ss-DsRed. These data are now included in new panels in Figure 4 and S10.

If not the Golgi, what do the perinuclear regions indicated represent, and where are the cargos transported to?

To unambiguously identify the perinuclear compartment where cargoes accumulate following release during prolonged nutrient depletion, we carried out immunostaining and found that ManII-SBP-GFP and ss-DsRed colocalized with Golgi membranes labeled by GM130 (see new Figure S7).

5, At the end of the paper, the authors attempted to demonstrate that a gain-of-function in Sec23a assembly rate could overcome cargo trafficking deficits. However, I did not see any evidence of artificially increased Sec23a assembly rate in Supplementary Figure 8a-b, where the duration and assembly events of Sec23a were unchanged.

We apologize for any confusion that we created with regard to this comment. By revising our set of figures, we hope that we have made our points more clear. In particular, we have revised Figure 6 to more clearly show the effect of overexpressing Sec23b on the formation rate of HaloTag-Sec23a structures. Notably, the increased formation rate demonstrated by overexpressing GFP-Sec23b is very similar to that found at 24 hours of nutrient depletion, as compared to 2 hours of nutrient depletion (please compare Figure 6B and 6C specifically).

Moreover, in Figure 6e, the authors showed that GFP-Sec23b R722A had a deficit in cargo transport compared to WT-Sec23b, which left me puzzled about the logic behind the claim. It would be more precise if the authors included a GAP activity gain-of-function mutant and a vector control in their experiments. Additionally, while the authors used ss-DsRed as a cargo indicator in Figure 6e, what about ManII? Only when the cargos move normally from the ER to the Golgi can it be considered "bypassing cargo trafficking deficits." Or it could be somehow an artificial phenomenon.

With regard to use of Sec23b (R722A), we agree that this convolutes our manuscript, and we have chosen not to include data relevant to this loss-of-function mutant. With respect to ManII-SBP-GFP, long-term nutrient deprivation failed to restore its rate of trafficking to that observed under nutrient replete conditions, suggesting that elevating HaloTag-Sec23a recruitment would be unlikely to expedite its rate of transport. Instead, we now examine another cargo beyond ss-DsRed (SNAP-tag-ERGIC-53), which is a native cargo of COPII in cells. During short-term nutrient limitation, increasing HaloTag-Sec23a recruitment was sufficient to restore SNAP-tag-ERGIC-53 accumulation (see Figure S10). These results are consistent with our findings using ss-DsRed and importantly do not rely on overexpression of an artificial cargo.

6, The author underscores the potential bias of the expression system at the outset, opting for the endogenous system. However, in Figure 6, the author resorts to the overexpression system to clarify some phenomena, which presents a degree of self-contradiction.

While we agree with the reviewer that the overexpression of GFP-Sec23b introduces caveats,

we are unaware of an alternative strategy to increase HaloTag-Sec23a recruitment to ER subdomains. Thus, for this single portion of the manuscript, we were forced to use an overexpression system, although all of our data analysis focuses on effects of endogenously expressed HaloTag-Sec23a.

Minor points:

1, The description of Figures 4b and 4c appears to be missing in the main text.

We apologize for this oversight, which has been corrected in the revised version of our manuscript.

2, The authors first mentioned the Sec23b R722A mutant as a GAP activity loss-of-function mutant, stating: "To determine whether the GAP activity of Sec23 is necessary for this effect, we overexpressed a mutant form of GFP-Sec23b, which harbors the R722A mutation that lacks GAP activity in vitro (49)." They also cited a reference (49) by Zacharogianni et al. in EMBO Journal 2011. However, I could not find any description of Sec23b GAP activity or the R722A mutant in the paper. Are the authors trying to emphasize a different point here?

We apologize for this error. However, all data related to Sec23b R722A have been removed from the manuscript, as suggested by the reviewer.

Reviewer #2 (Remarks to the Author):

In this paper, Kasberg et al. examined dynamics of COPII assembly and secretory cargo export from the ER and the effects of nutrient deprivation on these processes. By live imaging, they found that the rate of inner COPII coat assembly is important for determining the pace of cargo export rate. They also showed that the increase of inner coat assembly kinetics is sufficient to rescue cargo export deficiency caused by acute nutrient limitation.

This is a very interesting piece of work. I am impressed by how carefully the authors designed the experimental system to avoid possible artifacts caused by expression of tagged proteins. The expression of HaloTag-Sec16a appeared to be a little hazardous but the authors honestly described it. The effects of nutrient starvation are very complicated, but the authors' interpretations appear to be reasonable, although I am not 100% convinced that they are the only possible explanations.

We appreciate these comments and hope that the additional data included in our revised manuscript further convince the reviewer of the interpretations reached.

The results of this kind of detailed analysis on COPII assembly in vivo should be valuable for researchers in this field and therefore I am basically supportive for the publication of this work. However, I would like to ask the authors to address the following comments and questions before the final acceptance for publication.

1. Trafficking of ss-DsRed is compromised by 2-h nutrient depletion but recovers after 24 h. However, that of MnlI-SBP-GFP appears to be more defective at 24 h. On the other hand, L1CAM shows even higher efficiency of transport after 24-h nutrient depletion. How do the authors explain these differences?

Please see our response to reviewer 1 (specifically, to comment #4). Also, please note that we now examine the trafficking of an endogenous cargo, avoiding the use of artificial cargo release

systems, which have not been fully characterized.

2. The authors observed GRASP65 to reveal the Golgi morphology, saying that it is involved in cisternal stacking. However, there are arguments on the roles of GRASP proteins and the authors should be careful in the interpretation of the results.

We apologize for this error and now examine GM130, in addition to GRASP65, under conditions of nutrient limitation.

3. The authors argue the effect of GFP-Sec23b by comparing two different figures (e.g. Fig. 5e and 6b, Fig. 3b and 6c). This is not fair. Readers cannot evaluate it objectively.

We agree with the reviewer and have revised Figure 6 to address this concern.

Reviewer #3 (Remarks to the Author):

The manuscript "Nutrient deprivation alters the rate of COPII coat assembly to tune secretory protein transport" by Kasberg et al. describes changes in several COPII and COPII-related dynamics resulting from nutrient deprivation that culminates in affecting transport rates. I apologize in advance for sounding harsh. In my humble opinion, this manuscript suffers from several problematic issues, both technical and scientific. I will address a few technical issues, although they usually belong to the "minor comments" part.

We are disappointed that we failed to meet this reviewer's expectations, in contrast to the relatively positive comments provided by reviewers 1 and 2. We hope that our revisions will be more well received, which include substantial new data and extensive text revisions to try and address concerns raised.

Most of the references are provided in clusters making a follow-up almost impossible. Also, it almost seems that this is a means for the authors to avoid potentially controversial issues, and in so doing, they missed vital manuscripts that would have helped in describing at least some of the data presented. I will be more specific later in this review.

We apologize for the approach we used to cite prior work. It was certainly not our goal to avoid controversial issues. In the revised manuscript, citations are now provided mid-sentence in many cases to avoid clusters of references.

All figures are not numbered, making the review process annoying.

We upload each figure with a figure number individually, per journal policy. It is also our understanding that figure numbers should not be embedded in figures, so we are unsure how to satisfy the reviewer in this context.

Also, the figure legend lacks information to the extent that while writing this review, in some cases, I did not understand what the experiments performed based on the figures.

We apologize for not providing sufficient information in figure legends. We have now tried to extend explanations and added more details to all figure legends, without including interpretations since that would not be appropriate.

In some cases, the authors send the reader to a reference w/o even briefly describing the

experimental details.

We have now tried to include more experimental details throughout the manuscript, including the methods section, which includes numerous experimental details.

Moreover, while using many types of state-of-the-art equipment, the authors use the term quantitative while very few numbers are presented. Many graphs can be fitted to various simple forms of exponential equations, and the rate constant (Sec⁻¹) or inverse (Sec) may provide hard numbers for comparing the effect of the various treatments.

We have now tried to include more numbers throughout the manuscript, as requested by the reviewer. In particular, we have analyzed the formation of structures harboring HaloTag fusion proteins more extensively, including the extraction of rate constants that are now included. The revised figures are now more quantitative in nature as well.

A significant issue is the term "assembly," used numerous times throughout this manuscript. Unless I did not understand the data presented, the authors seem not to understand that COPII components dynamically bind to membranes meaning that the accurate description provided should be on/off rates.

We apologize for the mis-use of the term "assembly". This has been replaced throughout the manuscript with more accurate terminology.

I also fail to understand the data as, from this reviewer's experience, the "ER subdomains" known for many decades as transitional ER or better ER exit sites are long-lived at the scale of at least tens of minutes.

The reviewer is correct to point out that 'ER exit sites' have historically been described to be very long-lived, on the order of tens of minutes. As we now state in the text, many of the structures harboring HaloTag fusion proteins are indeed long-lived. However, we also identify many short-lived sites (ie., exist for less than 10 minutes). As described in our response to reviewer #1, long-lived sites correlate with increased intensity and size, suggesting that these structures are actually composed of multiple budding events that appear as single sites when imaged using diffraction-limited microscopy. Our interest was to leverage tools to track less intense, more dynamic sites over time, which has not been feasible previously due to limitations in technology. We believe that leveraging lattice light-sheet microscopy has revealed a more complete spectrum of behaviors exhibited by COPII proteins and their regulators, particularly those that are more transient in nature. This conclusion is further supported by the frequencies of different site lifetimes being best modeled by both fast and slow time constants. If all 'ER exit sites' lasted for tens of minutes, their lifetimes would likely be best modeled as single-component exponentials, which we found not to be the case. It is also important to note that previous studies have largely relied on the overexpression of COPII and COPII-associated proteins, which could extend their lifetimes artificially (DOI: 10.1242/jcs.113.12.2177; please see in particular the abnormally sized sites shown in figures 1 and 2). For these reasons, we believe that a focus on shorter-lived sites is critical to comprehend how COPII and its regulators are recruited dynamically to facilitate cargo export from the ER.

Relying only on the RUSH system may pose a problem as regardless of its advantages, it uses an artificial retention mechanism entirely unrelated to the natural ones. They should consider using the thermoreversible folding mutant of Vesicular Stomatitis virus G or collagen.

We agree with the reviewer, but also wish to point out that there are shortcomings associated with the use of thermoreversible VSVG and collagen, both of which are typically overexpressed to study their trafficking. We instead carried out HILO imaging of cells engineered to natively express ERGIC-53 fused to SNAP-tag, which represents a well-characterized client of COPII-mediated transport. By measuring de novo accumulation of endogenous SNAP-tag-ERGIC-53, we circumvent the need to use artificial synchronized release assays. Using this approach, we demonstrated that SNAP-tag-ERGIC-53 accumulates more quickly under both nutrient replete conditions and prolonged nutrient deprivation, as compared to more acute nutrient depletion.

There are other issues to address, but I will end by directing the authors to two manuscripts they cite buried in clusters, one in Cell (39) and the other in JCB (40), providing a new model for the localization and mechanism of function of COPII. This is also summarized in a recent review 1. In my opinion, this new dogma for how the early secretory works may very elegantly explain the major findings of this manuscript.

We agree with the reviewer that our findings are in agreement with recent studies that suggest COPII acts as a collar at membrane 'necks'. However, the prior work did not examine how nutrient availability influences COPII subunit recruitment, nor demonstrate that the local concentration of COPII subunits controls the rate of ER cargo transport, which we believe are the major new findings of our manuscript.

On a final positive note, the finding that rapid coat on/off dynamics promotes a faster transport rate is potentially very interesting as it may provide mechanistic information on how COPII selects and sorts cargo proteins.

We entirely agree with the reviewer on this point.

Reference:

1. Malis, Y., Hirschberg, K., and Kaether, C. (2022). Hanging the coat on a collar: Same function but different localization and mechanism for COPII. *Bioessays* 44, e2200064. [10.1002/bies.202200064](https://doi.org/10.1002/bies.202200064).

Please note that we reference this publication specifically in our revised manuscript.

REVIEWER COMMENTS

Reviewer #1 (Remarks to the Author):

The authors have addressed my concerns

Reviewer #2 (Remarks to the Author):

The manuscript looks improved. I am satisfied by the revisions made.

Reviewer #3 (Remarks to the Author):

I have many issues with the wording the authors choose. Especially after I have informed the authors of a new dogma regarding How and where COPII is functioning. This was not for the purpose of citing one or two manuscripts but my intention was for the authors to assimilate the new dogma in their manuscript. Thus, my impression is that at best they stayed "sitting on the fence" more leaning toward the older dogma. After reading the revised manuscript carefully, I am sorry to say that in my opinion, this manuscript is too vague in defining what exactly are they looking at. Indeed, state-of-the-art microscopy combined with sophisticated genome editing but the experiments are not well defined or explained. Also, my very long experience with the quantification of various aspects of ER exit sites, their sorting dynamics, and various other kinetics does not prevent me from failing to understand what are they essentially seeing. All four COPII components form a collar on the ER-ER exit site boundary. The ER exit sites are stable structures that are not even disrupted using intracellular transport inhibitors. All of these proteins dynamically bind to the collar so their dynamics should be described by on/off rates probably using some form of FRAP experiments.

I therefore conclude that this paper does not contribute to our understanding of the regulation of COPII during starvation. Below are specific issues that contributed to my decision to reject this manuscript:

"Co-assembly of the multilayered coat protein complex II (COPII) with the Sar1 GTPase at subdomains of the endoplasmic reticulum (ER) enables secretory cargoes to be concentrated efficiently within nascent transport intermediates,"

Cargo is essentially concentrated in ER exit sites whose membrane is still continuous with the ER. Describing the COPII accumulation sites as defined throughout as “ER-subdomains” is not very informative and essentially renders the entire manuscript vague as to the nature of this type of site. Are they ER exit sites? Transport carriers? Intermediate compartment? In the literature, ER-subdomains are mentioned in the context of ER-lipid droplet contact sites.

I also do not accept that the stability of ER exit sites is a result of the over-expression of COPII components. In one of the manuscripts they cite, Stable HeLa cells expressing a CRISPR/CAS12 knock-in of Sec13-mCherry were used. A time-lapse video shows that most if not all ER exit sites were apparently stable for over 40 min.

Did the authors rule out that the unstable “subdomains” were an artifact of the Halo tag?

The authors focused on Sec23 as a representative of the Inner coat. However, Sec24 is at least as interesting as it is the subunit that forms the interface with the cargo.

The reply to my 1st review that VSVG or collagen are problematic as they are overexpressed is simply wrong. The cargo molecules used by the authors are also overexpressed and use a completely artificial retention mechanism. Also, quantitative analysis of VSVG secretion after overnight accumulation in the ER for example was shown to not saturate secretory transport throughout the entire pathway. The RUSH system is too artificial and is in close spatial and temporal proximity to be used in a study that attempts to quantify cargo export from the ER.

The following phrase completely ignores the new dogma positioning COPII at the ER-ER exit site boundary ruling out its traditional function as a vesicle or carrier coat.

...” These intermediates undergo maturation and subsequently deliver their contents to distinct ER-Golgi intermediate compartments (ERGIC) (38-40) or an interwoven tubular network that is connected to the ER (41-43), which is facilitated by the Sec23-binding protein TFG (44-46).”

The term ss-DsRed is misleading as it is ss-DsRed-FKBP. The signal sequence is cleaved upon entry into the ER lumen and the remaining DsRed is considered a soluble protein lacking any export signal. Its ER export may be by passively entering the ER Exit site lumen

This is essentially an unsubstantiated conclusion from an observation of “ER subdomains” blinking out:

“...the longer-lived structures were generally larger and more intense, potentially representing multiple, closely juxtaposed COPII budding events...”

Maybe I missed it but what is the percentage of unstable “ER subdomains” compared to total “ER subdomains”?

The authors should be aware that ERGIC53 is part of the secretory machinery and by all means is not a cargo protein although considered as such occasionally.

I think that there is no such word as Cargoes

Rab1 was found to be a major player in regulating early secretory transport in health and disease(s) and is not even mentioned here.

Response to Reviewers' comments:

Reviewer #1 (Remarks to the Author):

The authors have addressed my concerns.

We are pleased that we have addressed all concerns raised previously by this reviewer.

Reviewer #2 (Remarks to the Author):

The manuscript looks improved. I am satisfied by the revisions made.

We are again pleased that we have addressed all concerns raised previously by this reviewer.

Reviewer #3 (Remarks to the Author):

I have many issues with the wording the authors choose. Especially after I have informed the authors of a new dogma regarding How and where COPII is functioning. This was not for the purpose of citing one or two manuscripts but my intention was for the authors to assimilate the new dogma in their manuscript. Thus, my impression is that at best they stayed “sitting on the fence” more leaning toward the older dogma.

This comment is highly concerning and seems to indicate a clear bias on the part of the reviewer. Although it is unclear, we believe the reviewer may be referring to the findings of a single manuscript published two years ago as ‘new dogma’ (PMCID: PMC8054201). Although that study suggests COPII may not coat transport carriers, there are decades of work indicating otherwise. Moreover, this controversy is not relevant to our study, as we aim to define how nutrient availability influences cargo trafficking and COPII accumulation at sites where transport carriers form. ‘Assimilating’ a single viewpoint into our manuscript would incorporate a major bias, which is neither appropriate nor reasonable, given the focus of our work.

After reading the revised manuscript carefully, I am sorry to say that in my opinion, this manuscript is too vague in defining what exactly are they looking at. Indeed, state-of-the-art microscopy combined with sophisticated genome editing but the experiments are not well defined or explained. Also, my very long experience with the quantification of various aspects of ER exit sites, their sorting dynamics, and various other kinetics does not prevent me from failing to understand what are they essentially seeing. All four COPII components form a collar on the ER-ER exit site boundary. The ER exit sites are stable structures that are not even disrupted using intracellular transport inhibitors. All of these proteins dynamically bind to the collar so their dynamics should be described by on/off rates probably using some form of FRAP experiments.

These comments are also very concerning and again show a high degree of bias on the part of the reviewer. The idea that COPII components form a collar is an opinion of the reviewer, but there are decades of published evidence to indicate the contrary. The idea that ER exit sites are stable structures has not been directly examined nor demonstrated. To our knowledge, our study is one

of the first to address this concept using high speed imaging of natively-tagged COPII components, and the first to use single particle tracking and dynamic fluorescence measurements to follow all COPII-labeled sites in cells. The use of FRAP would not address the questions we are posing, as FRAP is a method for determining the kinetics of diffusion, not measuring the accumulation of COPII subunits at individual sites within cells.

I therefore conclude that this paper does not contribute to our understanding of the regulation of COPII during starvation. Below are specific issues that contributed to my decision to reject this manuscript: “Co-assembly of the multilayered coat protein complex II (COPII) with the Sar1 GTPase at subdomains of the endoplasmic reticulum (ER) enables secretory cargoes to be concentrated efficiently within nascent transport intermediates,” Cargo is essentially concentrated in ER exit sites whose membrane is still continuous with the ER. Describing the COPII accumulation sites are defined throughout as “ER-subdomains” is not very informative and essentially renders the entire manuscript vague as to the nature of this type of site. Are they ER exit sites? Transport carriers? Intermediate compartment? In the literature, ER-subdomains are mentioned in the context of ER-lipid droplet contact sites.

In 1975, George Palade first defined ribosome-free ER subdomains that contain protrusions resembling budding vesicles (PMID: 1096303), which he referred to as transitional elements of the ER. These subdomains are very well defined, unlike the term ‘ER exit site’ or ‘ERES’, which has been used interchangeably for many years to refer to these ribosome-free ER subdomains as well as the entire interface between the ER membrane and ER-Golgi intermediate compartments (commonly known as ERGIC). Given the diffraction limits of lattice light-sheet microscopy and confocal microscopy, it is not possible to indicate whether the fluorescent, COPII-labeled structures visualized are transport carriers or an intermediate compartment. To do so would be an extreme overinterpretation of our data. Instead, using the term ‘ER subdomain’ seems most appropriate. Nonetheless, to ensure clarity, we have revised the manuscript text within the introduction to clearly state the definition of the term ‘ER subdomain’.

I also do not accept that the stability of ER exit sites is a result of the over-expression of COPII components. In one of the manuscripts they cite, Stable HeLa cells expressing a CRISPR/CAS12 knock-in of Sec13-mCherry were used. A time-lapse video shows that most if not all ER exit sites were apparently stable for over 40 min.

The time-lapse video noted by the reviewer uses single plane imaging, with images acquired every 15 seconds, which does not allow for analysis of the dynamics of sites labeled by Sec13-mCherry. Sites clearly go in and out of focus, or undergo disassembly during the imaging sequence. Based on these data, no conclusion can be reached regarding the stability of these sites. Instead, our approach leveraging high speed lattice light-sheet imaging, which captures full cell volumes at ~3 second time resolution, is the only approach currently available to analyze the formation of COPII-labeled sites in living cells, at least to our knowledge. Based on these studies, we find that COPII-labeled sites are not as stable as the reviewer postulates.

Did the authors rule out that the unstable “subdomains” were an artifact of the Halo tag?

We have systematically demonstrated the functionality of the HaloTag fusion proteins used in our study. Additionally, we previously published studies examining the dynamics of other proteins fused to HaloTag, none of which exhibited artifacts, as suggested by the reviewer. Moreover, in a separate manuscript currently in preparation, we analyzed the dynamics of a GFP fusion to Sec31a and found it behaves identically to HaloTag-Sec31a used in this study. Together, these data strongly argue against the possibility of artifacts arising from the use of the HaloTag.

The authors focused on Sec23 as a representative of the Inner coat. However, Sec24 is at least as interesting as it is the subunit that forms the interface with the cargo.

Unlike Sec23, which has two isoforms in mammalian cells that largely overlap in function, there are four Sec24 isoforms, which bind to distinct cargoes and cannot substitute for one another. These isoforms may exhibit completely distinct dynamics, and thus fail to provide a good representation of inner COPII coat dynamics. Thus, we have chosen to examine Sec23.

The reply to my 1st review that VSVG or collagen are problematic as they are overexpressed is simply wrong. The cargo molecules used by the authors are also overexpressed and use a completely artificial retention mechanism. Also, quantitative analysis of VSVG secretion after overnight accumulation in the ER for example was shown to not saturate secretory transport throughout the entire pathway. The RUSH system is too artificial and is in close spatial and temporal proximity to be used in a study that attempts to quantify cargo export from the ER.

The first publication describing the 'RUSH' system has been cited 483 times since its initial description in 2012 and in our opinion has been invaluable to the field to allow investigators to quantify cargo export from the ER. Nonetheless, the reviewer is correct that the system is artificial, similar to overexpressing a mutant form of VSV-G that misfolds at elevated temperature in the ER lumen. Thus, we also examine a native COPII cargo (ERGIC-53), expressed at endogenous levels to validate all of our results obtained using the RUSH system and another artificial release system (ss-DsRed). Thus, instead of relying on results obtained examining a single cargo, we study multiple to help ensure the rigor of our findings.

The following phrase completely ignores the new dogma positioning COPII at the ER-ER exit site boundary ruling out its traditional function as a vesicle or carrier coat.

..." These intermediates undergo maturation and subsequently deliver their contents to distinct ER-Golgi intermediate compartments (ERGIC) (38-40) or an interwoven tubular network that is connected to the ER (41-43), which is facilitated by the Sec23-binding protein TFG (44-46)."

With respect, the results from the study cited as 'the new dogma' do not rule out the traditional function of COPII as a coat. The statement we include in our manuscript is an unbiased assessment of nine rigorous studies published over the last ~30 years.

The term ss-DsRed is misleading as it is ss-DsRed-FKBP. The signal sequence is cleaved upon entry into the ER lumen and the remaining DsRed is considered a soluble protein lacking any export signal. Its ER export may be by passively entering the ER Exit site lumen.

As described in the study we cite in our manuscript (PMCID: PMC7927198), ss-DsRed is fused to a tripeptide, which is recognized by the Erv29 cargo receptor. This confers rapid transport from the ER to the Golgi. Thus, its ER export is not mediated by passive entry into the secretory pathway, based on prior published work.

This is essentially an unsubstantiated conclusion from an observation of “ER subdomains” blinking out:

“...the longer-lived structures were generally larger and more intense, potentially representing multiple, closely juxtaposed COPII budding events...”

Maybe I missed it but what is the percentage of unstable “ER subdomains” compared to total “ER subdomains”?

The statement highlighted is a potential explanation of our finding that longer-lived structures are generally larger and more intense. It is not a ‘conclusion’, as suggested by the reviewer. Supplementary Figure 5 includes histograms, illustrating the frequency of sites with differing longevities.

The authors should be aware that ERGIC53 is part of the secretory machinery and by all means is not a cargo protein although considered as such occasionally.

With respect, the reviewer is incorrect. ERGIC-53 has been demonstrated to be a cargo of the COPII machinery, with crystallographic evidence showing the interface between it and the Sec24a cargo receptor (PMCID: PMC5464768). Additionally, in reconstitution studies, ERGIC-53 is one of the most commonly examined cargoes in COPII budding assays originally developed by the Schekman lab (ie., PMCID: PMC6589570).

I think that there is no such word as Cargoes.

Based on a PubMed search, the term ‘cargoes’ has been used more than 25,000 times in the titles of manuscripts published previously.

Rab1 was found to be a major player in regulating early secretory transport in health and disease(s) and is not even mentioned here.

Rab1 has been shown to localize to ERGIC membranes and the cis-Golgi, but not on the ER in mammalian cells (PMID: 7615674). Thus, including Rab1 in our study of COPII dynamics at ER subdomains would be inappropriate.

REVIEWERS' COMMENTS

Reviewer #4 (Remarks to the Author):

I have been asked to comment on the controversy between the authors (A) and Reviewer 3 (R).

After having read the paper I comment below on each point. Overall I believe this is a valuable paper which contains novel information and is interpreted in a balanced manner. The reviewer raises some interesting points some of which the authors take on board, but also seems to be negatively biased and fails to provide much constructive criticism or propose how the authors could interpret their data in the context of new models.

R) I have many issues with the wording the authors choose. Especially after I have informed the authors of a new dogma regarding How and where COPII is functioning. This was not for the purpose of citing one or two manuscripts but my intention was for the authors to assimilate the new dogma in their manuscript. Thus, my impression is that at best they stayed “sitting on the fence” more leaning toward the older dogma.

A) This comment is highly concerning and seems to indicate a clear bias on the part of the reviewer. Although it is unclear, we believe the reviewer may be referring to the findings of a single manuscript published two years ago as ‘new dogma’ (PMCID: PMC8054201). Although that study suggests COPII may not coat transport carriers, there are decades of work indicating otherwise. Moreover, this controversy is not relevant to our study, as we aim to define how nutrient availability influences cargo trafficking and COPII accumulation at sites where transport carriers form. ‘Assimilating’ a single viewpoint into our manuscript would incorporate a major bias, which is neither appropriate nor reasonable, given the focus of our work.

I) I agree with the authors here. The new model proposed in PMCID: PMC8054201 is interesting and suggest that regulation of membrane remodelling at ER exit sites in animals might be a complex affair indeed. However, this model does not at all constitute a ‘new dogma’, as I believe the claim that COPII does not form vesicular carriers is not supported by the data in that paper. But in any case, I also agree that the authors are not attempting to resolve whether COPII subunits are retained as a stable coat on individual COPII vesicles, but are looking at average bulk recruitment at ERES, so the reviewer’s concern is ill-posed.

R) After reading the revised manuscript carefully, I am sorry to say that in my opinion, this manuscript is too vague in defining what exactly are they looking at. Indeed, state-of-the-art microscopy combined

with sophisticated genome editing but the experiments are not well defined or explained. Also, my very long experience with the quantification of various aspects of ER exit sites, their sorting dynamics, and various other kinetics does not prevent me from failing to understand what are they essentially seeing. All four COPII components form a collar on the ER-ER exit site boundary. The ER exit sites are stable structures that are not even disrupted using intracellular transport inhibitors. All of these proteins dynamically bind to the collar so their dynamics should be described by on/off rates probably using some form of FRAP experiments.

A) These comments are also very concerning and again show a high degree of bias on the part of the reviewer. The idea that COPII components form a collar is an opinion of the reviewer, but there are decades of published evidence to indicate the contrary. The idea that ER exit sites are stable structures has not been directly examined nor demonstrated. To our knowledge, our study is one of the first to address this concept using high speed imaging of natively-tagged COPII components, and the first to use single particle tracking and dynamic fluorescence measurements to follow all COPII-labeled sites in cells. The use of FRAP would not address the questions we are posing, as FRAP is a method for determining the kinetics of diffusion, not measuring the accumulation of COPII subunits at individual sites within cells.

I) I agree with the authors: I have not seen any data indicating that COPII forms collars. This is true for COPII-interacting proteins such as Tango1, which is not an object of this study at all. It is true what reviewer 3 says, that ERES are stable structures, but they do not persist forever. For the ERES that appear and disappear within the time frame of data collection, the authors here measure dynamics of bulk recruitment of COPII subunits showing that various coat components differ from each other and that differences arise in conditions of low nutrient availability.

R) I therefore conclude that this paper does not contribute to our understanding of the regulation of COPII during starvation. Below are specific issues that contributed to my decision to reject this manuscript: "Co-assembly of the multilayered coat protein complex II (COPII) with the Sar1 GTPase at subdomains of the endoplasmic reticulum (ER) enables secretory cargoes to be concentrated efficiently within nascent transport intermediates," Cargo is essentially concentrated in ER exit sites whose membrane is still continuous with the ER. Describing the COPII accumulation sites are defined throughout as "ER-subdomains" is not very informative and essentially renders the entire manuscript vague as to the nature of this type of site. Are they ER exit sites? Transport carriers? Intermediate compartment? In the literature, ER-subdomains are mentioned in the context of ER-lipid droplet contact sites.

A) In 1975, George Palade first defined ribosome-free ER subdomains that contain protrusions resembling budding vesicles (PMID: 1096303), which he referred to as transitional elements of the ER. These subdomains are very well defined, unlike the term 'ER exit site' or 'ERES', which has been used interchangeably for many years to refer to these ribosome-free ER subdomains as well as the entire

interface between the ER membrane and ER-Golgi intermediate compartments (commonly known as ERGIC). Given the diffraction limits of lattice light-sheet microscopy and confocal microscopy, it is not possible to indicate whether the fluorescent, COPII-labeled structures visualized are transport carriers or an intermediate compartment. To do so would be an extreme overinterpretation of our data. Instead, using the term 'ER subdomain' seems most appropriate. Nonetheless, to ensure clarity, we have revised the manuscript text within the introduction to clearly state the definition of the term 'ER subdomain'.

I) I do not see a problem here or grounds for rejection based on a definition of 'ER subdomain'. The authors' approach to define it is the correct way to settle the controversy.

R) I also do not accept that the stability of ER exit sites is a result of the over-expression of COPII components. In one of the manuscripts they cite, Stable HeLa cells expressing a CRISPR/CAS12 knock-in of Sec13-mCherry were used. A time-lapse video shows that most if not all ER exit sites were apparently stable for over 40 min.

A) The time-lapse video noted by the reviewer uses single plane imaging, with images acquired every 15 seconds, which does not allow for analysis of the dynamics of sites labeled by Sec13-mCherry. Sites clearly go in and out of focus, or undergo disassembly during the imaging sequence. Based on these data, no conclusion can be reached regarding the stability of these sites. Instead, our approach leveraging high speed lattice light-sheet imaging, which captures full cell volumes at ~3 second time resolution, is the only approach currently available to analyze the formation of COPII-labeled sites in living cells, at least to our knowledge. Based on these studies, we find that COPII-labeled sites are not as stable as the reviewer postulates.

R) Did the authors rule out that the unstable "subdomains" were an artifact of the Halo tag?

A) We have systematically demonstrated the functionality of the HaloTag fusion proteins used in our study. Additionally, we previously published studies examining the dynamics of other proteins fused to HaloTag, none of which exhibited artifacts, as suggested by the reviewer. Moreover, in a separate manuscript currently in preparation, we analyzed the dynamics of a GFP fusion to Sec31a and found it behaves identically to HaloTag-Sec31a used in this study. Together, these data strongly argue against the possibility of artifacts arising from the use of the HaloTag.

I) I agree that the authors cannot absolutely exclude the halo tag has any effects on the dynamics/stability of the subdomains. However, I think there is strong support for the fact that these effects would be minimal, and not of the extent that they would change persistence by orders of magnitude. It should be noted they measure different kinetics with different COPII subunits, but all of

them are in the same range and it would be hard to imagine how the halo tag addition would cause exactly the same artifact on the assembly stability of all components.

R) The authors focused on Sec23 as a representative of the Inner coat. However, Sec24 is at least as interesting as it is the subunit that forms the interface with the cargo.

A) Unlike Sec23, which has two isoforms in mammalian cells that largely overlap in function, there are four Sec24 isoforms, which bind to distinct cargoes and cannot substitute for one another. These isoforms may exhibit completely distinct dynamics, and thus fail to provide a good representation of inner COPII coat dynamics. Thus, we have chosen to examine Sec23.

I) I think it is unreasonable to expect that experiments looking at Sec24 should be included here, although a sentence commenting on the fact that its dynamics might be different due to cargo-mediated retention and might be paralogue-specific could be added.

R) The reply to my 1st review that VSVG or collagen are problematic as they are overexpressed is simply wrong. The cargo molecules used by the authors are also overexpressed and use a completely artificial retention mechanism. Also, quantitative analysis of VSVG secretion after overnight accumulation in the ER for example was shown to not saturate secretory transport throughout the entire pathway. The RUSH system is too artificial and is in close spatial and temporal proximity to be used in a study that attempts to quantify cargo export from the ER.

A) The first publication describing the 'RUSH' system has been cited 483 times since its initial description in 2012 and in our opinion has been invaluable to the field to allow investigators to quantify cargo export from the ER. Nonetheless, the reviewer is correct that the system is artificial, similar to overexpressing a mutant form of VSV-G that misfolds at elevated temperature in the ER lumen. Thus, we also examine a native COPII cargo (ERGIC-53), expressed at endogenous levels to validate all of our results obtained using the RUSH system and another artificial release system (ss-DsRed). Thus, instead of relying on results obtained examining a single cargo, we study multiple to help ensure the rigor of our findings.

I) I think the authors went to great length here to satisfy the reviewer's (legitimate) concern that RUSH and overexpression might both give rise to non-physiological observations.

R) The following phrase completely ignores the new dogma positioning COPII at the ER-ER exit site boundary ruling out its traditional function as a vesicle or carrier coat.

..." These intermediates undergo maturation and subsequently deliver their contents to distinct ER-Golgi intermediate compartments (ERGIC) (38-40) or an interwoven tubular network that is connected to the ER (41-43), which is facilitated by the Sec23-binding protein TFG (44-46)."

A) With respect, the results from the study cited as 'the new dogma' do not rule out the traditional function of COPII as a coat. The statement we include in our manuscript is an unbiased assessment of nine rigorous studies published over the last ~30 years.

I) I totally agree with the authors here, as I write above, the 'new dogma' is to my opinion based on over-interpretation of a single observation. While it does challenge the simplistic nature of the 'classic' vesicular model, it would be wholly premature and inappropriate to discard vesicle-mediated transport as the main mode of ER exit (whether COPII is stably bound to such vesicles or not).

R) The term ss-DsRed is misleading as it is ss-DsRed-FKBP. The signal sequence is cleaved upon entry into the ER lumen and the remaining DsRed is considered a soluble protein lacking any export signal. Its ER export may be by passively entering the ER Exit site lumen.

A) As described in the study we cite in our manuscript (PMCID: PMC7927198), ss-DsRed is fused to a tripeptide, which is recognized by the Erv29 cargo receptor. This confers rapid transport from the ER to the Golgi. Thus, its ER export is not mediated by passive entry into the secretory pathway, based on prior published work.

I) I have no comment on this

R) This is essentially an unsubstantiated conclusion from an observation of "ER subdomains" blinking out:

"...the longer-lived structures were generally larger and more intense, potentially representing multiple, closely juxtaposed COPII budding events..."

Maybe I missed it but what is the percentage of unstable "ER subdomains" compared to total "ER subdomains"?

A) The statement highlighted is a potential explanation of our finding that longer-lived structures are generally larger and more intense. It is not a 'conclusion', as suggested by the reviewer. Supplementary Figure 5 includes histograms, illustrating the frequency of sites with differing longevities.

I) I do not understand the reviewer's concern here. Nevertheless, the authors do say they measured kinetics for subdomains for which they could detect a start and end, and it would be interesting to quantitate their number compared to subdomains that live longer than the timeframe of the experiments (and were not measured)

R) The authors should be aware that ERGIC53 is part of the secretory machinery and by all means is not a cargo protein although considered as such occasionally.

A) With respect, the reviewer is incorrect. ERGIC-53 has been demonstrated to be a cargo of the COPII machinery, with crystallographic evidence showing the interface between it and the Sec24a cargo receptor (PMCID: PMC5464768). Additionally, in reconstitution studies, ERGIC-53 is one of the most commonly examined cargoes in COPII budding assays originally developed by the Schekman lab (ie., PMCID: PMC6589570).

I) I agree with the authors, many cargoes are part of the secretory machinery (e.g. SNARES). These proteins are transported out of the ER via COPII, and in this sense they are COPII cargoes.

R) I think that there is no such word as Cargoes.

A) Based on a PubMed search, the term 'cargoes' has been used more than 25,000 times in the titles of manuscripts published previously.

I) Both cargos and cargoes are used and suggested as correct options by most English dictionaries.

R) Rab1 was found to be a major player in regulating early secretory transport in health and disease(s) and is not even mentioned here.

A) Rab1 has been shown to localize to ERGIC membranes and the cis-Golgi, but not on the ER in mammalian cells (PMID: 7615674). Thus, including Rab1 in our study of COPII dynamics at ER subdomains would be inappropriate.

I) I believe the reviewer refers to this paper: PMC7390636. The authors are right that Rab1 is described to act at non-ER associated sites, therefore not relevant to include as a target for this study. However, the

reviewer has a point in that this paper should be acknowledged in the discussion as it contains data relevant to the persistence of COPII protein at ER exit sites.

Response to Reviewers' comments:

Reviewer #4 (Remarks to the Author):

I have been asked to comment on the controversy between the authors (A) and Reviewer 3 (R). After having read the paper I comment below on each point. Overall I believe this is a valuable paper which contains novel information and is interpreted in a balanced manner. The reviewer raises some interesting points some of which the authors take on board, but also seems to be negatively biased and fails to provide much constructive criticism or propose how the authors could interpret their data in the context of new models.

We appreciate Reviewer 4's perspective.

R) I have many issues with the wording the authors choose. Especially after I have informed the authors of a new dogma regarding How and where COPII is functioning. This was not for the purpose of citing one or two manuscripts but my intention was for the authors to assimilate the new dogma in their manuscript. Thus, my impression is that at best they stayed "sitting on the fence" more leaning toward the older dogma.

A) This comment is highly concerning and seems to indicate a clear bias on the part of the reviewer. Although it is unclear, we believe the reviewer may be referring to the findings of a single manuscript published two years ago as 'new dogma' (PMCID: PMC8054201). Although that study suggests COPII may not coat transport carriers, there are decades of work indicating otherwise. Moreover, this controversy is not relevant to our study, as we aim to define how nutrient availability influences cargo trafficking and COPII accumulation at sites where transport carriers form. 'Assimilating' a single viewpoint into our manuscript would incorporate a major bias, which is neither appropriate nor reasonable, given the focus of our work.

I) I agree with the authors here. The new model proposed in PMCID: PMC8054201 is interesting and suggest that regulation of membrane remodelling at ER exit sites in animals might be a complex affair indeed. However, this model does not at all constitute a 'new dogma', as I believe the claim that COPII does not form vesicular carriers is not supported by the data in that paper. But in any case, I also agree that the authors are not attempting to resolve whether COPII subunits are retained as a stable coat on individual COPII vesicles, but are looking at average bulk recruitment at ERES, so the reviewer's concern is ill-posed.

We appreciate that Reviewer 4 agrees with our perspective.

R) After reading the revised manuscript carefully, I am sorry to say that in my opinion, this manuscript is too vague in defining what exactly are they looking at. Indeed, state-of-the-art microscopy combined with sophisticated genome editing but the experiments are not well defined or explained. Also, my very long experience with the quantification of various aspects of ER exit sites, their sorting dynamics, and various other kinetics does not prevent me from failing to understand what are they essentially seeing. All four COPII components form a collar on the ER-ER exit site boundary. The ER exit sites are stable structures that are not even disrupted using intracellular transport inhibitors. All of these proteins dynamically bind to the collar so

their dynamics should be described by on/off rates probably using some form of FRAP experiments.

A) These comments are also very concerning and again show a high degree of bias on the part of the reviewer. The idea that COPII components form a collar is an opinion of the reviewer, but there are decades of published evidence to indicate the contrary. The idea that ER exit sites are stable structures has not been directly examined nor demonstrated. To our knowledge, our study is one of the first to address this concept using high speed imaging of natively-tagged COPII components, and the first to use single particle tracking and dynamic fluorescence measurements to follow all COPII-labeled sites in cells. The use of FRAP would not address the questions we are posing, as FRAP is a method for determining the kinetics of diffusion, not measuring the accumulation of COPII subunits at individual sites within cells.

I) I agree with the authors: I have not seen any data indicating that COPII forms collars. This is true for COPII-interacting proteins such as Tango1, which is not an object of this study at all. It is true what reviewer 3 says, that ERES are stable structures, but they do not persist forever. For the ERES that appear and disappear within the time frame of data collection, the authors here measure dynamics of bulk recruitment of COPII subunits showing that various coat components differ from each other and that differences arise in conditions of low nutrient availability.

We appreciate that Reviewer 4 agrees with our perspective.

R) I therefore conclude that this paper does not contribute to our understanding of the regulation of COPII during starvation. Below are specific issues that contributed to my decision to reject this manuscript: “Co-assembly of the multilayered coat protein complex II (COPII) with the Sar1 GTPase at subdomains of the endoplasmic reticulum (ER) enables secretory cargoes to be concentrated efficiently within nascent transport intermediates,” Cargo is essentially concentrated in ER exit sites whose membrane is still continuous with the ER. Describing the COPII accumulation sites are defined throughout as “ER-subdomains” is not very informative and essentially renders the entire manuscript vague as to the nature of this type of site. Are they ER exit sites? Transport carriers? Intermediate compartment? In the literature, ER-subdomains are mentioned in the context of ER-lipid droplet contact sites.

A) In 1975, George Palade first defined ribosome-free ER subdomains that contain protrusions resembling budding vesicles (PMID: 1096303), which he referred to as transitional elements of the ER. These subdomains are very well defined, unlike the term ‘ER exit site’ or ‘ERES’, which has been used interchangeably for many years to refer to these ribosome-free ER subdomains as well as the entire interface between the ER membrane and ER-Golgi intermediate compartments (commonly known as ERGIC). Given the diffraction limits of lattice light-sheet microscopy and confocal microscopy, it is not possible to indicate whether the fluorescent, COPII-labeled structures visualized are transport carriers or an intermediate compartment. To do so would be an extreme overinterpretation of our data. Instead, using the term ‘ER subdomain’ seems most appropriate. Nonetheless, to ensure clarity, we have revised the manuscript text within the introduction to clearly state the definition of the term ‘ER subdomain’.

I) I do not see a problem here or grounds for rejection based on a definition of ‘ER subdomain’. The authors’ approach to define it is the correct way to settle the controversy.

We appreciate that Reviewer 4 agrees with our perspective.

R) I also do not accept that the stability of ER exit sites is a result of the over-expression of COPII components. In one of the manuscripts they cite, Stable HeLa cells expressing a CRISPR/CAS12 knock-in of Sec13-mCherry were used. A time-lapse video shows that most if not all ER exit sites were apparently stable for over 40 min.

A) The time-lapse video noted by the reviewer uses single plane imaging, with images acquired every 15 seconds, which does not allow for analysis of the dynamics of sites labeled by Sec13-mCherry. Sites clearly go in and out of focus, or undergo disassembly during the imaging sequence. Based on these data, no conclusion can be reached regarding the stability of these sites. Instead, our approach leveraging high speed lattice light-sheet imaging, which captures full cell volumes at ~3 second time resolution, is the only approach currently available to analyze the formation of COPII-labeled sites in living cells, at least to our knowledge. Based on these studies, we find that COPII-labeled sites are not as stable as the reviewer postulates.

Reviewer 4 did not comment specifically on this point. We continue to believe that COPII-labeled sites are not as stable as Reviewer 3 postulates.

R) Did the authors rule out that the unstable “subdomains” were an artifact of the Halo tag?

A) We have systematically demonstrated the functionality of the HaloTag fusion proteins used in our study. Additionally, we previously published studies examining the dynamics of other proteins fused to HaloTag, none of which exhibited artifacts, as suggested by the reviewer. Moreover, in a separate manuscript currently in preparation, we analyzed the dynamics of a GFP fusion to Sec31a and found it behaves identically to HaloTag-Sec31a used in this study. Together, these data strongly argue against the possibility of artifacts arising from the use of the HaloTag.

D) I agree that the authors cannot absolutely exclude the halo tag has any effects on the dynamics/stability of the subdomains. However, I think there is strong support for the fact that these effects would be minimal, and not of the extent that they would change persistence by orders of magnitude. It should be noted they measure different kinetics with different COPII subunits, but all of them are in the same range and it would be hard to imagine how the halo tag addition would cause exactly the same artifact on the assembly stability of all components.

We appreciate that Reviewer 4 agrees with our perspective.

R) The authors focused on Sec23 as a representative of the Inner coat. However, Sec24 is at least as interesting as it is the subunit that forms the interface with the cargo.

A) Unlike Sec23, which has two isoforms in mammalian cells that largely overlap in function, there are four Sec24 isoforms, which bind to distinct cargoes and cannot substitute for one

another. These isoforms may exhibit completely distinct dynamics, and thus fail to provide a good representation of inner COPII coat dynamics. Thus, we have chosen to examine Sec23.

I) I think it is unreasonable to expect that experiments looking at Sec24 should be included here, although a sentence commenting on the fact that its dynamics might be different due to cargo-mediated retention and might be paralogue-specific could be added.

We have added a comment as suggested in the discussion section.

R) The reply to my 1st review that VSVG or collagen are problematic as they are overexpressed is simply wrong. The cargo molecules used by the authors are also overexpressed and use a completely artificial retention mechanism. Also, quantitative analysis of VSVG secretion after overnight accumulation in the ER for example was shown to not saturate secretory transport throughout the entire pathway. The RUSH system is too artificial and is in close spatial and temporal proximity to be used in a study that attempts to quantify cargo export from the ER.

A) The first publication describing the ‘RUSH’ system has been cited 483 times since its initial description in 2012 and in our opinion has been invaluable to the field to allow investigators to quantify cargo export from the ER. Nonetheless, the reviewer is correct that the system is artificial, similar to overexpressing a mutant form of VSV-G that misfolds at elevated temperature in the ER lumen. Thus, we also examine a native COPII cargo (ERGIC-53), expressed at endogenous levels to validate all of our results obtained using the RUSH system and another artificial release system (ss-DsRed). Thus, instead of relying on results obtained examining a single cargo, we study multiple to help ensure the rigor of our findings.

I) I think the authors went to great length here to satisfy the reviewer’s (legitimate) concern that RUSH and overexpression might both give rise to non-physiological observations.

We appreciate that Reviewer 4 agrees with our perspective.

R) The following phrase completely ignores the new dogma positioning COPII at the ER-ER exit site boundary ruling out its traditional function as a vesicle or carrier coat.

...” These intermediates undergo maturation and subsequently deliver their contents to distinct ER-Golgi intermediate compartments (ERGIC) (38-40) or an interwoven tubular network that is connected to the ER (41-43), which is facilitated by the Sec23-binding protein TFG (44-46).”

A) With respect, the results from the study cited as ‘the new dogma’ do not rule out the traditional function of COPII as a coat. The statement we include in our manuscript is an unbiased assessment of nine rigorous studies published over the last ~30 years.

I) I totally agree with the authors here, as I write above, the ‘new dogma’ is to my opinion based on over-interpretation of a single observation. While it does challenge the simplistic nature of the ‘classic’ vesicular model, it would be wholly premature and inappropriate to discard vesicle-mediated transport as the main mode of ER exit (whether COPII is stably bound to such vesicles or not).

We appreciate that Reviewer 4 agrees with our perspective.

R) The term ss-DsRed is misleading as it is ss-DsRed-FKBP. The signal sequence is cleaved upon entry into the ER lumen and the remaining DsRed is considered a soluble protein lacking any export signal. Its ER export may be by passively entering the ER Exit site lumen.

A) As described in the study we cite in our manuscript (PMCID: PMC7927198), ss-DsRed is fused to a tripeptide, which is recognized by the Erv29 cargo receptor. This confers rapid transport from the ER to the Golgi. Thus, its ER export is not mediated by passive entry into the secretory pathway, based on prior published work.

I) I have no comment on this

Although Reviewer 4 has no comment on this, we continue to believe that ss-DsRed export from the ER is not mediated by passive entry into the secretory pathway.

R) This is essentially an unsubstantiated conclusion from an observation of “ER subdomains” blinking out:

“...the longer-lived structures were generally larger and more intense, potentially representing multiple, closely juxtaposed COPII budding events...”

Maybe I missed it but what is the percentage of unstable “ER subdomains” compared to total “ER subdomains”?

A) The statement highlighted is a potential explanation of our finding that longer-lived structures are generally larger and more intense. It is not a ‘conclusion’, as suggested by the reviewer. Supplementary Figure 5 includes histograms, illustrating the frequency of sites with differing longevities.

I) I do not understand the reviewer’s concern here. Nevertheless, the authors do say they measured kinetics for subdomains for which they could detect a start and end, and it would be interesting to quantitate their number compared to subdomains that live longer than the timeframe of the experiments (and were not measured)

We appreciate Reviewer 4’s comment. As we stated in response to Reviewer 3, the data in Supplementary Figure 5 highlight the frequency of sites with differing longevities.

R) The authors should be aware that ERGIC53 is part of the secretory machinery and by all means is not a cargo protein although considered as such occasionally.

A) With respect, the reviewer is incorrect. ERGIC-53 has been demonstrated to be a cargo of the COPII machinery, with crystallographic evidence showing the interface between it and the Sec24a cargo receptor (PMCID: PMC5464768). Additionally, in reconstitution studies, ERGIC-53 is one of the most commonly examined cargoes in COPII budding assays originally developed by the Schekman lab (ie., PMCID: PMC6589570).

I) I agree with the authors, many cargoes are part of the secretory machinery (e.g. SNARES). These proteins are transported out of the ER via COPII, and in this sense they are COPII cargoes.

We appreciate that Reviewer 4 agrees with our perspective.

R) I think that there is no such word as Cargoes.

A) Based on a PubMed search, the term ‘cargoes’ has been used more than 25,000 times in the titles of manuscripts published previously.

I) Both cargos and cargoes are used and suggested as correct options by most English dictionaries.

We appreciate Reviewer 4’s comment and continue to use the term ‘cargoes’ in our manuscript.

R) Rab1 was found to be a major player in regulating early secretory transport in health and disease(s) and is not even mentioned here.

A) Rab1 has been shown to localize to ERGIC membranes and the cis-Golgi, but not on the ER in mammalian cells (PMID: 7615674). Thus, including Rab1 in our study of COPII dynamics at ER subdomains would be inappropriate.

I) I believe the reviewer refers to this paper: PMC7390636. The authors are right that Rab1 is described to act at non-ER associated sites, therefore not relevant to include as a target for this study. However, the reviewer has a point in that this paper should be acknowledged in the discussion as it contains data relevant to the persistence of COPII protein at ER exit sites. The authors have addressed my concerns.

We have added a citation to PMC7390636, as suggested. We are pleased that we have addressed Reviewer 4’s concerns.